# LEARNPRUNER: RETHINKING ATTENTION-BASED TOKEN PRUNING IN VISION LANGUAGE MODELS

**Rinyoichi Takezoe, Yaqian Li, Zihao Bo, Anzhou Hou, Mo Guang, Kaiwen Long**[*]

Li Auto Inc.

`longkaiwen@lixiang.com`

## ABSTRACT

Vision-Language Models (VLMs) have recently demonstrated remarkable capabilities in visual understanding and reasoning, but they also impose significant computational burdens due to long visual sequence inputs. Recent works address this issue by pruning unimportant visual tokens, achieving substantial computational reduction while maintaining model performance. The core of token pruning lies in determining token importance, with current approaches primarily relying on attention scores from vision encoders or Large Language Models (LLMs). In this paper, we analyze the effectiveness of attention mechanisms in both vision encoders and LLMs. We find that vision encoders suffer from attention sink, leading to poor focus on informative foreground regions, while in LLMs, although prior studies have identified attention bias toward token positions, text-to-vision attention demonstrates resistance to this bias and enables effective pruning guidance in middle layers. Based on these observations, we propose **LearnPruner**, a two-stage token pruning framework that first removes redundant vision tokens via a learnable pruning module after the vision encoder, then retains only task-relevant tokens in the LLM's middle layer. Experimental results show that our LearnPruner can preserve approximately 95% of the original performance while using only 5.5% of vision tokens, and achieve $3.2\times$ inference acceleration, demonstrating a superior accuracy-efficiency trade-off.

## 1 INTRODUCTION

In recent years, with the rapid advancement of Large Language Models (LLMs) Guo et al. (2025); Achiam et al. (2023); Bai et al. (2023), Vision-Language Models (VLMs) have achieved remarkable progress. By extending the comprehension and reasoning capabilities of LLMs to the visual modality, VLMs have demonstrated unprecedented performance across various multimodal tasks, including visual question answering, image captioning, and visual reasoning Li et al. (2023a); Liu et al. (2023); Wang et al. (2024).

Existing VLMs typically convert images into discrete token sequences through vision encoders and then feed them into LLMs after modal alignment modules. However, the massive number of visual tokens poses significant computational challenges for VLM inference, particularly in high-resolution image or long-video input scenarios. This computational burden severely limits the practical deployment of VLMs in resource-constrained environments and real-time applications.

To mitigate these computational challenges, token pruning has emerged as a promising solution. The core idea of token pruning is to assign importance scores to individual visual tokens through specific criteria, then keep only the top-k most important tokens while discarding the rest during the inference phase. Since visual content typically has much lower information density than text, substantial reductions in visual token count through pruning techniques yield minimal performance degradation.

Obviously, the effectiveness of token pruning heavily depends on accurate visual token importance assessment. Recent studies primarily rely on attention scores from visual encoders or LLMs as

---
[*]Corresponding author.

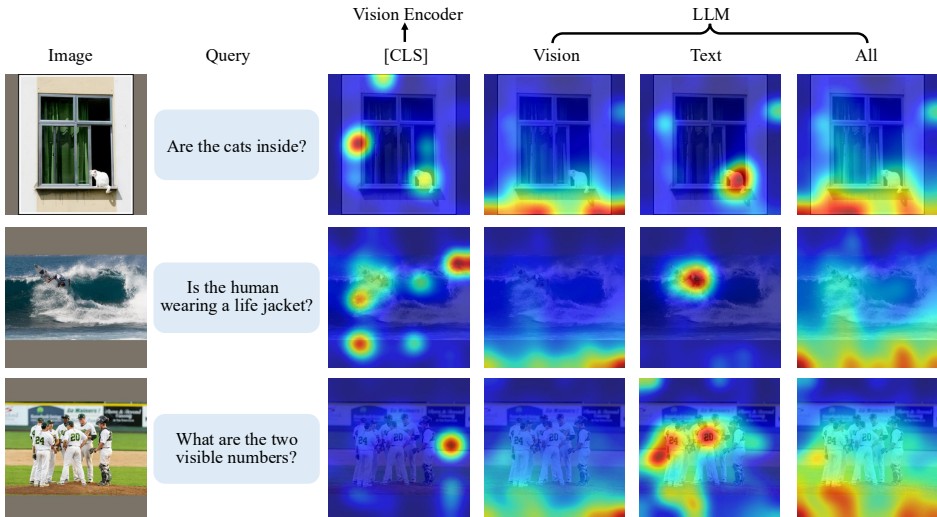

Figure 1: Attention maps of vision encoder and LLM decoder in LLaVA-1.5 Liu et al. (2023). The former leverages the attention scores of the `[CLS]` token derived from the second last layer of the vision encoder, while the latter shows the average attention received by vision tokens from vision, text, and all tokens in the middle layer (12th layer) of the LLM.

the metrics, e.g., `[CLS]` attention Zhang et al. (2025a); Yang et al. (2025) or average LLM attention Chen et al. (2024); Zhang et al. (2025b). To investigate whether attention mechanisms accurately reflect token importance, we visualize the attention heatmaps of LLaVA-1.5 Liu et al. (2023), a widely-used VLM, from both the vision encoder and LLM.

As shown in Figure 1, while the `[CLS]` token can partially focus on foreground objects, it often allocates excessive attention to low-informative background regions. This observation is consistent with the prior work Darcet et al. (2023) that shows Vision Transformers Oquab et al. (2023); Radford et al. (2021) tend to generate high-norm outlier tokens, i.e., artifacts that aggregate global image information while discarding spatial detail. On the other hand, while prior work Zhang et al. (2025a) identified the attention shift phenomenon in LLMs, where LLMs exhibit a bias toward the lower half of images due to the locality of positional encoding and causal masking mechanisms, this bias is mainly observed in vision-to-vision or all-token attention patterns. In contrast, text-to-vision attention appears to effectively focus on query-relevant regions.

The above observations motivate us to conduct an in-depth analysis on the effectiveness of current attention-based token pruning methods, revealing two critical insights: (1) In vision encoder, `[CLS]` token fails to adequately attend to salient foreground objects, which leads to suboptimal pruning results, particularly under limited token budgets. (2) In LLM, text-to-vision attention demonstrates robustness in resisting attention shift, and can provide reliable guidance for selecting query-relevant vision tokens in the middle layer. However, delaying pruning until the middle layers still involves substantial redundant computations, resulting in marginal acceleration gains.

To address these limitations, we propose **LearnPruner**, a two-stage token pruning framework that enhances the efficiency of VLMs by sequentially removing unnecessary vision tokens after the vision encoder and in the LLM. Specifically, we first eliminate inherent visual redundancy by employing a lightweight learnable module to directly predict the importance scores of vision tokens, replacing the conventional `[CLS]` attention scores. Moreover, a small set of diversity tokens are retained to provide complementary visual information. The remaining tokens then pass into the LLM for cross-modal interaction with the query instructions. Subsequently, we perform query-aware token selection in the middle layer of LLM, further discarding tokens that are irrelevant to the given query.

Benefiting from the refined pruning strategy and importance measures, LearnPruner achieves a favorable accuracy-efficiency trade-off. Extensive experiments on various VLM benchmarks demonstrate that LearnPruner outperforms previous state-of-the-art methods. With only 5.6% of tokens

retained, LearnPruner preserves 94.8% of the original performance and achieves a $2.3\times$ and $1.5\times$ speedup in prefill and total time, respectively.

## 2 RELATED WORK

### 2.1 VISION-LANGUAGE MODELS.

The evolution of Vision-Language Models (VLMs) has progressed from early joint embedding approaches Kiros et al. (2014); Karpathy & Fei-Fei (2015) to sophisticated architectures leveraging Large Language Model capabilities. Modern VLMs such as LLaVA-1.5 Liu et al. (2023), MiniGPT-4 Zhu et al. (2023), and Qwen-VL Bai et al. (2023) have demonstrated remarkable performance in multimodal understanding by integrating visual encoders with powerful language models through modal alignment modules.

Moreover, recent VLM inputs have expanded to high-resolution images and video sequences, dramatically increasing visual token sequences. For example, LLaVA-1.5 Liu et al. (2023) generates 576 tokens per image, LLaVA-Next Liu et al. (2024a) divides the high-resolution images into a grid of sub-images, generating up to $5 \times 576 = 2,880$ tokens, and LLaVA-OneVision Li et al. (2024) applies pooling operations to video frames, creating up to $32 \times 196 = 6,272$ tokens. This trend toward longer visual sequences has not only increased computational burden but also created a significant imbalance in multimodal inputs, where visual tokens often dominate the input composition. However, visual modalities exhibit substantially higher redundancy compared to language, leading to a mismatch between token quantity and information density, which motivates the necessity for visual token reduction research in VLMs.

### 2.2 VISUAL TOKEN REDUCTION FOR VLMS.

As mentioned above, visual token reduction aims to improve inference efficiency by removing redundant information in images. FastV Chen et al. (2024) is a pioneering work in this field, which computes the average attention scores one token received from all other tokens within the LLM to determine token importance, and prunes unimportant tokens at the shallow layer of the LLM. PyramidDrop Xing et al. (2024) observes that redundancy increases progressively within LLMs, thus proposing a hierarchical pruning strategy. SparseVLM Zhang et al. (2025b) leverages text-to-vision attention scores to achieve text-aware guidance, and utilizes the rank of attention matrices to adaptively adjust the pruning ratio. Meanwhile, some studies apply pruning strategies directly after the vision encoder. VisPruner Zhang et al. (2025a) argues that attention shift and attention dispersion issues exist within LLMs, therefore replaces the importance criteria with `[CLS]` token attention scores. VisionZip Yang et al. (2025) further incorporates the token merging technique to avoid losing any small but potentially important information. Alternatively, from a diversity perspective, DivPrune Alvar et al. (2025) and DART Wen et al. (2025) select tokens based on feature similarity to maintain a diverse set that captures comprehensive visual context.

In addition to the above training-free methods, recent studies have introduced training-based pruning methods to further improve accuracy, at the expanse of acceptable additional training overhead. ATP-LLaVA Ye et al. (2025) utilizes the global attention distribution of images to predict instance-specific pruning thresholds. TwigVLM Shao et al. (2025) inserts additional decoder blocks in the shallow layers of the LLM, which not only perform more reliable pruning, but also enable decoding stage acceleration through self-speculative decoding. Despite existing works relying on attention results to remove redundant tokens, the attention mechanism exhibits inherent limitations, and its effectiveness warrants further exploration.

## 3 METHOD

### 3.1 PRELIMINARY

Existing works typically rely on attention map derived from the transformer block to achieve token pruning. The attention mechanism Vaswani et al. (2017) is widely applied in both key components of VLMs (the vision encoder and the LLM) to facilitate token interaction. Formally, given a token

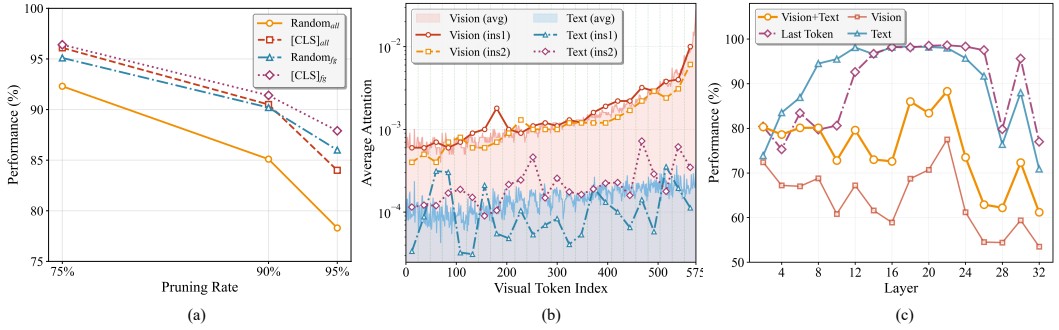

Figure 2: Analysis of attention in VLMs. (a) Performance comparison of token pruning strategies. 'Random' and '`[CLS]`' denote random and `[CLS]` attention-based selection from the entire image (all) or foreground regions (fg). (b) Distribution of attention received by vision tokens from both vision and text modalities. The bar chart presents the average values across multiple instances, while the dashed line chart shows the distribution of individual samples. (c) Performance comparison of importance criteria derived from different layers of LLM attention maps. Pruning is performed at the second layer with a 90% pruning ratio.

sequence $\mathbf{X} = [x_1, x_2, \cdots, x_N] \in \mathbb{R}^{N \times d}$, the attention computation first transforms the input into query $\mathbf{Q}$, key $\mathbf{K}$, and value $\mathbf{V}$ respectively:

$$\mathbf{Q} = \mathbf{X}\mathbf{W}_q, \quad \mathbf{K} = \mathbf{X}\mathbf{W}_k, \quad \mathbf{V} = \mathbf{X}\mathbf{W}_v \quad (1)$$

where $\mathbf{W}_q$, $\mathbf{W}_k$, and $\mathbf{W}_v$ are projection matrices, $N$ is the sequence length and $d$ is the hidden dimension. Then, the attention map $A$ and output $\mathbf{O}$ are computed as:

$$\mathbf{A}_{\text{logit}} = \mathbf{Q}\mathbf{K}^T, \quad \mathbf{A} = \text{softmax}\left(\frac{\mathbf{A}_{\text{logit}} + \mathbf{M}}{\sqrt{d}}\right), \quad \mathbf{O} = \mathbf{A}\mathbf{V} \quad (2)$$

where $\mathbf{A}$ denotes the attention map and $\mathbf{M} \in \mathbb{R}^{N \times N}$ is an optional mask matrix. For vision encoders adopting global attention, $\mathbf{M}$ is a zero matrix, while for LLMs employing causal attention, $\mathbf{M}$ is an upper triangular matrix of $-\infty$ values to ensure each token can only attend to previous tokens.

### 3.2 STUDY OF ATTENTION IN VLMS

**Attention in Vision Encoder.** Vision encoders commonly include an additional `[CLS]` token at the beginning of the token sequence, serving to interact with patch tokens and aggregate global information. The `[CLS]` token is expected to focus on the visually salient regions. Consequently, existing works naturally utilize the attention score from the `[CLS]` token as the importance estimation of the patch tokens. However, recent studies Darcet et al. (2023); Yang et al. (2024) reveal that most vision encoders tend to generate artifacts in uniform, background areas. These artifacts receive disproportionately high attention from other tokens, despite containing limited semantic information. Therefore a natural question arises: *whether the `[CLS]` token adequately attend to salient foreground regions?*

To answer this question, we design a comparative experiment using LLaVA-1.5-7B Liu et al. (2023) on five benchmarks: GQA Hudson & Manning (2019), POPE Li et al. (2023b), MME Fu et al. (2023), TextVQA Singh et al. (2019) and VQAv2 Goyal et al. (2017) (all subsequent experiments maintain the same setup unless otherwise specified). Specifically, we employ LangSAM[1], an open-source visual grounding tool built on top of SAM-2 Ravi et al. (2024) and GroundingDINO Liu et al. (2024b), to segment the foreground objects of the image. We pre-define a set of common foreground and background categories, and obtain the complete foreground mask based on the comprehensive segmentation results.

We then evaluate two pruning strategies: global token selection and foreground-constrained selection, the average performance is shown in Figure 2.(a). For `[CLS]` attention-based pruning, constraining token selection to informative regions (`[CLS]`$_{fg}$) consistently outperforms selection from

---

[1]https://github.com/luca-medeiros/lang-segment-anything

the entire image ($[\texttt{CLS}]_{all}$), especially under aggressive pruning ratios. Furthermore, even random foreground token selection ($\text{Random}_{fg}$) can achieve comparable performance to $[\texttt{CLS}]_{all}$. This suggests that the $[\texttt{CLS}]$ token may not effectively focus on salient foreground regions, particularly that the most attended tokens show poor alignment with their importance, which is consistent with the observation in Figure 1.

**Attention in LLM.** In LLMs, visual tokens not only interact within the modality but also with text tokens. Existing works typically utilize the average attention received by other tokens or the last instruction token to estimate the importance score, and pruning usually occurs in the shallow layers of LLMs to achieve lower computational costs. However, recent studies Zhang et al. (2025a) have identified the attention shift phenomenon in LLMs, where visual tokens with higher indices tend to receive higher attention scores. Intuitively, the information flow within LLMs should serve as guidance for token pruning, hence we ask: *Is there a more effective way to leverage LLM attention for pruning assistance?*

To answer this question, we first extend the analysis of attention shift based on Zhang et al. (2025a). We randomly select 1,000 image-text pairs from VQAv2 as inputs to the base VLM and compute the average attention received at each vision token index. The attention is decomposed into visual attention (from vision tokens) and text attention (from text tokens). As shown in Figure 2.(b), while both modalities demonstrates attention shift, the increasing trend in text attention is much more gradual than visual attention across token indices. This is

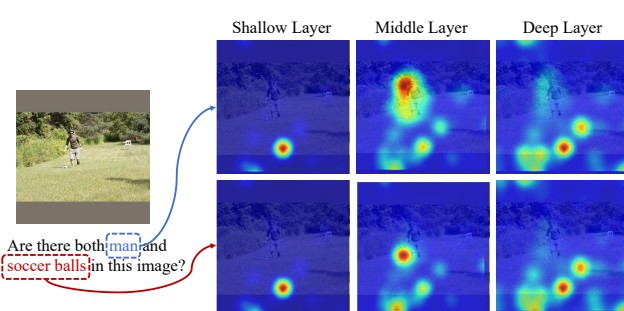

Figure 3: Attention heatmaps from specific text tokens to images in shallow, middle and deep layers of LLM.

mainly due to the decay property of positional encoding and the causal masking in visual attention. Meanwhile, although text attention overall exhibits a positive correlation with token index, attention patterns vary considerably across individual instances and tokens with lower indices can also be highly attended. In contrast, vision attention consistently shows bias, which may be harmful for pruning decisions.

Based on the observation, we further conduct experiments to compare different importance criteria within LLMs. The criteria include the average attention from visual modality, text modality, both modalities, as well as the attention from the last instruction token. For fair comparison of these criteria's effectiveness, we follow the approach of Shao et al. (2025) by pre-computing attention maps from a specific layer as guidance for token selection, then performing token pruning at the second layer. The pruning ratio is set to 90% and the results are shown in Figure 2.(c).

We can find that text attention and last token attention consistently outperform the other criteria across almost all layers. Incorporating visual attention leads to significant performance degradation, indicating that it is heavily affected by attention shift and fails to provide complementary information. Moreover, an interesting trend can be observed: attention reliability gradually increases from shallow layers (over 95% of the baseline performance is retained from layer 8 onwards) and shows stable performance in middle layers, but significantly declines as the layer goes deeper. A more concrete example is shown in Figure 3. In both shallow and deep layers, uninformative regions tend to absorb most of the attention from text tokens, while in the middle layer, different text tokens are able to focus on their semantic-related regions, hence allowing for effective token selection.

### 3.3 LEARNPRUNER

Based on the above study, we propose LearnPruner, which leverages learnable pruning criteria to replace the attention-based criteria in vision encoder and adopts a progressive pruning strategy, performing pruning after the vision encoder and within the LLM respectively. The overall framework is shown in Figure 4.

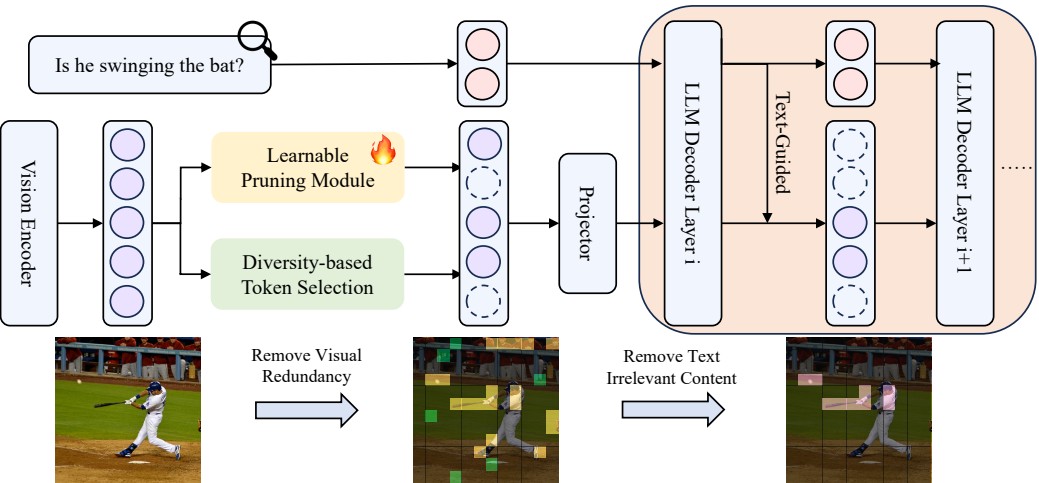

Figure 4: Overview of LearnPruner. After vision encoder, we select informative tokens based on importance socre predicted by learnable pruning module. Besides, a limited number of diverse tokens are selected based on similarity to minimize background information loss. Finally, we perform text-guided token selection in the middle layer of the LLM, further discarding tokens that are irrelevant to the given query.

**Remove Visual Redundancy.** Visual redundancy is inherently present in images, thus it is intuitive to perform pruning after the vision encoder. The stage aims to preserve the original image information through more compact token representations. Although `[CLS]` attention is widely used as the pruning criterion, as analyzed before, `[CLS]` token fails to accurately attend to visually salient regions. Inspired by Rao et al. (2021); Huang et al. (2024), we employ a learnable pruning module (LPM) to directly predict the importance score of each vision token.

Specifically, we feed the token features from the vision encoder into a lightweight MLP that performs binary classification to determine whether each token should be preserved or pruned. Since discrete binary decisions are non-differentiable, we employ Straight-Through Estimator (STE) Bengio et al. (2013) to enable backpropagation during training. This process can be formulated as:

$$\mathbf{M}_{\text{soft}} = \text{Softmax}(\text{MLP}(\mathbf{X}_v^{(0)})), \tag{3}$$

$$\mathbf{M}_{\text{hard}} = \text{argmax}(\mathbf{M}_{\text{soft}}), \tag{4}$$

where $\mathbf{X}_v^{(0)}$ denotes the token features from the vision encoder. Through the STE trick, the binary mask $\mathbf{M}_{\text{hard}}$ ensures that pruned tokens do not participate in subsequent computations during the forward pass, while the soft mask $\mathbf{M}_{\text{soft}}$ provides gradients during the backward pass. Implementation details of end-to-end training can be found in the supplementary material. At inference time, $\mathbf{M}_{\text{soft}}$ serves as the importance score for each token. More implementation details about LPM can be found in Appendix A.1.

Considering that LPM tends to focus on semantically rich foreground regions, it might ignore background information that could also be crucial in certain VQA tasks, therefore we introduce a diversity-based token selection module during inference to complement comprehensive visual context. After LPM selects a set of informative tokens, for each remaining token, we compute its cosine similarity with all selected tokens and identify its maximum similarity value. We then iteratively add the one with the smallest maximum similarity score, thereby ensuring maximum diversity in the selected set. This process continues until the token budget is reached.

**Remove Text-Irrelevant Content.** Although images contain vast information, not all visual content is necessary for the given query, hence we perform a second pruning within LLM to further remove text-irrelevant tokens. As demonstrated in our analysis, text attention is less affected by attention shift and exhibits strong response to relevant regions. Inspired by this, we directly leverage text attention to guide token pruning. The text attention is computed as the average attention from query

| Method | Params. | GQA | SQA$^I$ | VQA$^T$ | POPE | MME | VQA$^{v2}$ | MMB | MMB$^{CN}$ | RelAcc. |
|---|---|---|---|---|---|---|---|---|---|---|
| *Upper Bound, 576 Tokens (100%)* | | | | | | | | | | |
| Vanilla | - | 61.9 | 69.5 | 58.2 | 85.9 | 1862 | 78.5 | 64.7 | 58.3 | 100.0% |
| *Retain Averaged 128 Tokens (↓ 77.8%)* | | | | | | | | | | |
| FastV | - | 49.6 | 60.2 | 50.6 | 59.6 | 1490 | 61.8 | 56.1 | 51.4 | 82.1% |
| SparseVLM | - | 56 | 67.1 | 54.9 | 80.5 | 1696 | 73.8 | 60.0 | 51.1 | 92.6% |
| DivPrune | - | 59.3 | 69.0 | 56.1 | **86.7** | 1718 | 76.0 | 62.0 | 54.8 | 96.4% |
| DART | - | 57.9 | 69.1 | 56.3 | 80.4 | 1721 | 74.7 | 60.7 | **57.3** | 95.4% |
| VisPruner | - | 58.2 | 57.0 | 57.0 | 84.6 | 1794 | 75.8 | 62.7 | **57.3** | 97.3% |
| VisionZip | - | 57.6 | 68.9 | 56.8 | 83.2 | 1762 | 75.6 | 62.0 | 56.7 | 96.3% |
| VisionZip ‡ | 20.9M | 58.9 | 68.3 | 57.0 | 83.7 | **1823** | 76.6 | 62.6 | - | 97.3% |
| TwigVLM | 610M | **60.6** | **69.5** | **57.8** | 86.6 | 1818 | **77.9** | 63.5 | - | **99.0%** |
| LearnPruner | 0.53M | 60.3 | 68.5 | 57.3 | **86.7** | 1820 | 77.3 | **63.8** | 56.8 | 98.5% |
| *Retain Averaged 64 Tokens (↓ 88.9%)* | | | | | | | | | | |
| FastV | - | 46.1 | 51.1 | 47.8 | 48.0 | 1256 | 55.0 | 48.0 | 42.7 | 71.4% |
| SparseVLM | - | 52.7 | 62.2 | 51.8 | 75.1 | 1505 | 68.2 | 56.2 | 46.1 | 85.6% |
| DivPrune | - | 57.8 | 68.2 | 54.7 | 85.6 | 1674 | 74.1 | 59.3 | 52.3 | 93.9% |
| DART | - | 54.7 | 69.3 | 54.7 | 73.9 | 1705 | 71.3 | 59.5 | 54.0 | 91.9% |
| VisPruner | - | 55.4 | 69.1 | 55.8 | 80.4 | 1689 | 72.7 | 61.3 | 55.1 | 94.3% |
| VisionZip | - | 55.1 | 69.0 | 55.5 | 77.0 | 1690 | 72.4 | 60.1 | 55.4 | 93.0% |
| VisionZip ‡ | 20.9M | 57.0 | 68.8 | 56.0 | 80.9 | 1756 | 74.2 | 61.5 | - | 95.1% |
| TwigVLM | 610M | 58.8 | **70.0** | 55.8 | 82.7 | **1760** | 75.6 | 60.4 | - | 96.0% |
| LearnPruner | 0.53M | **58.9** | 68.3 | **56.6** | **86.8** | 1750 | **76.0** | **62.6** | **55.7** | **96.9%** |
| *Retain Averaged 32 Tokens (↓ 94.4%)* | | | | | | | | | | |
| FastV | - | 41.5 | 42.6 | 42.5 | 32.5 | 1090 | 43.4 | 37.8 | 33.2 | 58.6% |
| SparseVLM | - | 48.3 | 57.3 | 46.1 | 67.9 | 1290 | 58.6 | 51.4 | 40.6 | 77.5% |
| DivPrune | - | 54.9 | 68.6 | 52.9 | 81.5 | 1611 | 71.2 | 57.6 | 49.1 | 90.5% |
| DART | - | 52.9 | **69.3** | 52.2 | 69.1 | 1615 | 67.1 | 58.5 | 50.0 | 88.0% |
| VisPruner | - | 52.2 | 69.2 | 53.9 | 72.7 | 1567 | 67.7 | 58.4 | 52.7 | 89.7% |
| VisionZip | - | 51.8 | 68.8 | 53.1 | 68.7 | 1536 | 67.1 | 57.7 | 50.3 | 87.8% |
| LearnPruner | 0.53M | **57.2** | 68.2 | **56.1** | **84.5** | **1672** | **74.0** | **60.8** | **55.5** | **94.8%** |

Table 1: Performance comparisons of different pruning methods on LLaVA-v1.5-7B. 'Params.' denotes the number of learnable parameters and 'RelAcc.' denotes the average relative accuracy across all benchmarks compared to the vanilla model. Bold and underlined values indicate the best and second-best performance, respectively.

tokens over all heads:

$$\tilde{\mathbf{A}}^{(k)} = \frac{1}{N_q} \sum_{i=1}^{N_q} \mathbf{A}^{(k)}(\mathbf{X}_{q,i}^{(k)}, \mathbf{X}_v^{(k)}), \tag{5}$$

where $\mathbf{X}_q^{(k)}$ and $\mathbf{X}_v^{(k)}$ denote query tokens and vision tokens at the $k$-th layer, respectively. $N_q$ is the number of query tokens. We only keep the top-k tokens with the highest attention scores to participate in further interactions within the LLM.

## 4 EXPERIMENTS

In this section, we evaluate the performance of our LearnPruner on widely-used VLMs across various benchmarks, to compare with the state-of-the-art token pruning methods. Then we conduct ablation studies to analyze the effectiveness of each component.

### 4.1 EXPERIMENTAL SETUP

In the experiments, we train our model using 10% of the LLaVA-665K dataset Liu et al. (2023). We keep the base VLM weights frozen to preserve the original performance and only train the LPM. During inference, the first stage retains $R_1$ tokens after vision encoder, where $\lambda = 10\%$ of $R_1$ tokens are kept as diversity tokens. The second pruning stage is performed in the $k = 12$-th layer of the LLM, retaining $R_2$ tokens. For evaluation, we fix the average number of retained vision tokens

| Method | GQA | SQA[I] | VQA[T] | MME | VQA[v2] | MMB | RelAcc. |
|--------|-----|--------|--------|-----|---------|-----|---------|
| *Upper Bound, 2,880 Tokens (100%)* | | | | | | | |
| Vanilla | 64.2 | 70.2 | 61.3 | 1842 | 81.2 | 67.9 | 100% |
| *Retain Averaged 640 Tokens (↓ 77.8%)* | | | | | | | |
| SparseVLM | 60.3 | 67.7 | 57.8 | 1772 | 77.1 | 65.7 | 95.4% |
| DivPrune | 61.9 | 67.8 | 57.0 | 1760 | 79.3 | 65.8 | 96.0% |
| VisionZip | 61.3 | 68.1 | 60.2 | 1787 | 79.1 | 66.3 | 97.1% |
| VisionZip ‡ | 62.4 | 67.9 | **60.8** | 1778 | 79.9 | 65.9 | 97.5% |
| TwigVLM | 63.4 | 69.9 | 58.6 | **1864** | **81.2** | 67.4 | 99.1% |
| LearnPruner | **63.8** | **70.0** | 60.5 | 1837 | 80.5 | 67.5 | **99.3%** |
| *Retain Averaged 320 Tokens (↓ 88.9%)* | | | | | | | |
| SparseVLM | 57.7 | 67.3 | 55.9 | 1694 | 73.4 | 64.3 | 92.3% |
| DivPrune | 61.1 | 67.7 | 56.2 | 1724 | 77.2 | 63.9 | 94.3% |
| VisionZip | 59.3 | 67.3 | 58.9 | 1702 | 76.2 | 63.1 | 93.9% |
| VisionZip ‡ | 61.0 | 67.5 | **59.3** | 1770 | 78.4 | 64.4 | 95.9% |
| TwigVLM | **62.2** | **68.7** | 57.4 | 1758 | **79.7** | 65.0 | 96.3% |
| LearnPruner | **62.2** | 68.6 | 58.4 | **1845** | 78.3 | **66.8** | **97.5%** |
| *Retain Averaged 160 Tokens (↓ 94.4%)* | | | | | | | |
| SparseVLM | 51.2 | 67.5 | 46.4 | 1542 | 66.3 | 63.1 | 85.0% |
| DivPrune | **59.3** | 67.1 | 54.1 | 1643 | 75.0 | 62.9 | 91.7% |
| VisionZip | 55.5 | **68.3** | 56.2 | 1630 | 71.4 | 60.1 | 90.1% |
| VisionZip ‡ | 58.2 | 67.5 | **57.3** | 1699 | 75.6 | 63.9 | 93.3% |
| LearnPruner | 58.7 | 67.6 | 55.0 | **1784** | **76.2** | **65.3** | **94.0%** |

Table 2: Performance comparisons of different pruning methods on LLaVA-NeXT-7B.

across all LLM layers to ensure fair comparison. Given the target token budget, we set the ratio of $R_1$ and $R_2$ to 3 to determine their specific values.

| Method | TGIF-QA | | MSVD-QA | | MSRVTT-QA | | Average | |
|--------|---------|------|---------|------|-----------|------|---------|------|
| | Acc. | Score | Acc. | Score | Acc. | Score | Acc. | Score |
| *Upper Bound, 2,048 Tokens (100%)* | | | | | | | | |
| Vanilla | 18.1 | 1.27 | 64.1 | 3.41 | 56.1 | 2.97 | 46.1 | 2.55 |
| *Retain Averaged 456 Tokens (↓ 77.8%)* | | | | | | | | |
| FastV | 16.4 | 1.17 | 60.3 | 3.24 | 53.0 | 2.84 | 43.2 | 2.42 |
| LearnPruner | **17.5** | **1.25** | **63.6** | **3.36** | **55.3** | **2.93** | **45.5** | **2.51** |
| *Retain Averaged 228 Tokens (↓ 88.9%)* | | | | | | | | |
| FastV | 12.4 | 1.00 | 55.1 | 2.93 | 50.4 | 2.70 | 39.3 | 2.21 |
| LearnPruner | **16.3** | **1.17** | **61.5** | **3.30** | **53.8** | **2.87** | **43.9** | **2.45** |

Table 3: Video understanding performance comparisons of different pruning methods on Video-LLaVA-7B. Performance is evaluated on the first 1k samples of each benchmark. We use GPT-4.1 to assist in evaluating the accuracy.

## 4.2 MAIN RESULTS

To demonstrate the effectiveness of LearnPruner, we present a comprehensive evaluation of Learn-Pruner on various VLMs, including LLaVA-v1.5-7B, LLaVA-NeXT-7B for high resolution input, Video-LLaVA-7B for video input and Qwen2.5-VL-7B (beyond the LLaMA family).

**Results on LLaVA-v1.5-7B.** We first deploy our method on LLaVA-v1.5-7B, and then evaluate the performance on eight image understanding benchmarks. The compared methods include FastV Chen et al. (2024), SparseVLM Zhang et al. (2025b), DivPrune Alvar et al. (2025), DART Wen et al. (2025), VisionZip Yang et al. (2025), VisionZip ‡ Yang et al. (2025), Vis-Pruner Zhang et al. (2025a) and TwigVLM Shao et al. (2025). Among them, VisionZip‡ and TwigVLM require additional training, while the rest are training-free. As shown in Table 1, when the number of vision tokens is reduced from 576 to 128, LearnPruner only decreases the average accuracy by 1.5%, showing comparable performance to TwigVLM. It is worth noting that Learn-Pruner is more lightweight, as TwigVLM introduces additional decoder layers for training, leading to increased model complexity. Moreover, our method demonstrates greater advantages in scenarios

| Method | GQA | SQA$^I$ | VQA$^T$ | MME | VQA$^{v2}$ | MMB | Avg |
|---|---|---|---|---|---|---|---|
| *Upper Bound, 1280 Tokens (100%)* | | | | | | | |
| Vanilla | 59.7 | 77.4 | 76.7 | 2324 | 83.9 | 83.8 | 100% |
| *Retain Averaged 426 Tokens (↓ 66.7%)* | | | | | | | |
| FastV | 58.4 | **79.1** | **75.7** | 2330 | 81.4 | 81.4 | 98.9% |
| LearnPruner | **59.3** | 77.4 | 75.2 | **2339** | **82.4** | **83.2** | **99.3%** |
| *Retain Averaged 284 Tokens (↓ 77.8%)* | | | | | | | |
| FastV | 53.6 | **77.8** | 72.7 | 2228 | 77.8 | 79.1 | 94.7% |
| LearnPruner | **59.0** | 77.0 | **74.4** | **2358** | **81.7** | **82.2** | **98.7%** |
| *Retain Averaged 142 Tokens (↓ 88.9%)* | | | | | | | |
| FastV | 42.8 | 70.5 | 54.8 | 1565 | 57.2 | 56.2 | 71.4% |
| LearnPruner | **51.9** | **76.1** | **70.8** | **2292** | **79.3** | **81.8** | **94.1%** |

Table 4: Performance comparisons of different pruning methods on Qwen2.5-VL-7B.

| Method | Token | TFLOPs | Prefill Time | Total Time | KV Cache (MB) | GPU Memory (GB) |
|---|---|---|---|---|---|---|
| LLaVA-v1.5-7b | 576 | 8.6 (1.0×) | 463.2 (1.0×) | 761.6 (1.0×) | 318.9 (1.0×) | 14.7 (1.0×) |
| + LearnPruner | 128 | 2.8 (3.1×) | 262.6 (1.8×) | 569.8 (1.3×) | 95.0 (3.4×) | 13.5 (1.1×) |
| | 32 | 1.6 (5.4×) | 201.4 (2.3×) | 507.3 (1.5×) | 47.0 (6.8×) | 13.3 (1.1×) |
| LLaVA-NeXT-7b | 2880 | 31.9 (1.0×) | 1891.5 (1.0×) | 2307.6 (1.0×) | 1156.1 (1.0×) | 20.6 (1.0×) |
| + LearnPruner | 640 | 11.4 (2.8×) | 683.0 (2.8×) | 1083.6 (2.1×) | 369.4 (3.1×) | 16.3 (1.3×) |
| | 160 | 5.2 (6.1×) | 315.6 (6.0×) | 711.7 (3.2×) | 129.4 (8.9×) | 13.5 (1.5×) |

Table 5: Performance comparisons on computational efficiency.

with more constrained token budgets. With only 64 or 32 tokens retained, LearnPruner maintains 96.9% and 94.8% of the original performance, respectively, significantly outperforming other methods. This result suggests that our method can more effectively identify and preserve the most critical visual information.

**Results on LLaVA-NeXT-7B.** In Tabel 2, we further evaluate the performance of LearnPruner on LLaVA-NeXT-7B. LLaVA-NeXT splits the original image into multiple sub-images and feeds them together to the VLM to enable high-resolution input, resulting in vision token sequences that can reach up to 2880 tokens. Although LLaVA-NeXT improves the ability of image understanding, it also increases computational burden and introduces more visual redundancy. As we can observed, our LearnPruner consistently maintain strong performance under different settings. When removing 88.9% of tokens and retaining only 320 tokens, LearnPruner preserves 97.5% of the original performance.

**Results on Video-LLaVA-7B.** LearnPruner is a general pruning method that can be extended to video domain. We conduct experiments using Video-LLaVA-7B Lin et al. (2023) as the base model on three widely-used video understanding benchmarks: TGIF-QA Jang et al. (2017), MSVD-QA Xu et al. (2017), and MSRVTT-QA Xu et al. (2017). Following previous work Chen et al. (2024) and due to resource constraints, we evaluate on the first 1,000 samples from each benchmark and employ GPT Assistant to score the model responses. As shown in Table 3, LearnPruner achieves better accuracy and matching scores than FastV on all evaluated video QA tasks, further demonstrating the generalization across different application scenarios.

**Results on Qwen2.5-VL-7B.** To further demonstrate the generalization ability of LearnPruner, we conduct additional experiments on Qwen2.5-VL-7B Bai et al. (2025), a powerful VLM with a distinct architecture from the LLaVA series. We follow the same training settings and the results are shown in Table 4. LearnPruner still consistently outperforms FastV at different reduction ratios. When removing 88.9% of vision tokens, FastV's performance drops dramatically to 71.4% relative accuracy, while LearnPruner maintains 94.1% of the original performance, showing the superiority of our method.

## 4.3 ABLATION STUDIES

To validate the impact of different components in LearnPruner, we conduct ablation experiments on LLaVA-v1.5-7B. We evaluate the model on benchmarks mentioned in Table 1 and report the average performance, the number of retained tokens is fixed to 64. More ablation studies for pruning strategies are presented in Appendix A.2.

**Effectiveness of importance criteria.** The core of LearnPruner lies in replacing `[CLS]` attention-based importance criteria with learnable prediction scores, and implementing a two-stage pruning strategy. We conduct comparison experiments to verify the effectiveness of these design choices and the result is shown in Table 6. In the stage of removing visual redundancy, compared to selecting tokens by `[CLS]` attention, LPM improves performance by 1.7%, demonstrating that LPM can more effectively focus on salient foreground regions. Notably, using LPM alone in Stage 1 is conceptually similar to

| Stage1 | Stage2 | RelAcc.(%) |
|--------|--------|------------|
| `[CLS]` Attn | - | 94.6 |
| LPM | - | 96.1 |
| LPM | LPM | **96.9** |
|  | Text Attn | **96.9** |

Table 6: Effectiveness of different importance criteria.

Dynamic-LLaVA Huang et al. (2024), which also employs learned importance predictors for token selection. However, it can be seen that incorporating Stage2 further improves performance, demonstrating that our two-stage pruning strategy enables more effective token budget allocation. In the stage of removing text-irrelevant content, we attempt to insert an additional LPM module in the LLM as well, combining token features and attention distributions to predict importance scores. However, the results show no further performance improvement, indicating that attention signals are already sufficiently reliable and token features cannot provide complementary information. Therefore, we finally decide to directly utilize the attention results for pruning, avoiding the need to train multiple LPMs.

## 4.4 EFFICIENCY ANALYSIS

To demonstrate the efficiency of LearnPruner, we conduct a comparative analysis of computational cost and memory usage on LLaVA-v1.5-7B and LLaVA-Next-7B. As shown in Table 5, we use NVIDIA A100-80GB to perform the inference on POPE dataset. When the number of retained tokens is reduced from 576 to 32 in LLaVA-v1.5-7B, the prefill time and total time achieve $5.4\times$ and $2.3\times$ speedup respectively, while the KV cache storage is reduced by $6.8\times$. Due to our lightweight design, the computational overhead and memory usage of LPM are negligible. Moreover, the efficiency improvements become more pronounced as the visual token sequence length increases. When the number of retained tokens is reduced from 2,880 to 160 in LLaVA-Next-7B, the prefill time and total time achieve $6.0\times$ and $3.2\times$ speedup respectively.

## 5 CONCLUSION

In this paper, we conduct an in-depth analysis on attention mechanisms from both vision encoders and LLMs. Building on the observations, we propose LearnPruner, a two-stage pruning framework that first removes redundant vision tokens via a learnable pruning module after the vision encoder, then further discards text-irrelevant tokens in the LLM's middle layer. Extensive experimental results show that our LearnPruner outperforms previous state-of-the-art methods and achieves a better accuracy-efficiency trade-off.

## 6 ETHICS STATEMENT

The research presented in this paper focuses on token pruning for Vision-Language Models (VLMs) to improve their computational efficiency. The research process of this paper does not violate ICLR ethics. There are no discrimination, bias, or fairness issues that need to be addressed. Our models are not expected to generate potentially harmful content.

## 7 REPRODUCIBILITY STATEMENT

This article proposes a novel token pruning approach for VLMs to improve their inference efficiency. The base model and dataset used in this article are all from open-source and well-referenced, so this aspect does not affect the reproducibility. To further ensure reproducibility, we describe the implementation details in the main text Section 4.1 and the Appendix A.1. We will release the source code and model checkpoints to support reproducibility.

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

# A  APPENDIX

## A.1  TRAINING IMPLEMENTATION DETAILS

In this section, we introduce the training method of learnable pruning module in LearnPruner.

**End-to-end Optimization.** Unlike inference phase that we directly discard tokens based on the importance score $\mathbf{M}_{\text{soft}}$ predicted by LPM, we need to preserve the complete token sequence to optimize LPM in end to end training process.

Following Rao et al. (2021), given binary mask $\mathbf{M}_{\text{hard}}$, we cut off the interactions between pruned tokens and retained tokens in the attention layers to simulate the pruning process, enabling the LPM to learn how to assess token importance for the task. More specifically, we first obtain the complete token mask by substituting the visual token positions in an all-ones vector with the binary mask $\mathbf{M}_{\text{hard}}$ to create the full token sequence mask $\mathbf{M}$. Then we implement an attention masking strategy by modifying the Softmax$(\cdot)$ function:

$$
\mathbf{G}_{ij} = \begin{cases} 1, & i = j \\ \mathbf{M}_j, & i \neq j, \end{cases}
$$
$$
\hat{\mathbf{A}}_{ij} = \frac{\exp(\mathbf{Q}_i \mathbf{K}_j^T) \mathbf{G}_{ij}}{\sum_k \exp(\mathbf{Q}_i \mathbf{K}_k^T) \mathbf{G}_{ik}}. \tag{6}
$$

$\mathbf{G}$ is constructed as a graph adjacency matrix, where $\mathbf{G}_{ij} = 1$ denotes that the $j$-th token participates in the attention computation of the $i$-th token. With this design, $\hat{\mathbf{A}}$ is equivalent to the attention matrix calculated by actually discarding masked tokens.

**Objective Function.** Our goal is to encourage the model to use the least number of vision tokens to produce the correct answer. To achieve this, we introduce a pruning loss to constrain the number of kept tokens, which is defined as:

$$
\mathcal{L}_{\text{prune}} = \left( \frac{1}{N_v} \sum \mathbf{M}_{\text{hard}} - \tau \right)^2, \tag{7}
$$

where $N_v$ is the number of vision tokens and $\tau$ denotes the target ratio of kept tokens. We set $\tau = 0$ to encourage minimal token usage. With the regularization term, the final objective is:

$$
\mathcal{L} = \mathcal{L}_{\text{ntp}} + \lambda \cdot \mathcal{L}_{\text{prune}} \tag{8}
$$

where $\mathcal{L}_{\text{ntp}}$ is the original VLM loss and we set $\lambda = 1$ in all our experiments.

## A.2  MORE EXPERIMENTAL RESULTS

**Ablation on pruning position $k$.** The pruning position $k$ within the LLM affects the reliability of attention distribution and the number of tokens retained at each stage. Table 7a varies $k$ from 8 to 16 to investigate the impact on LearnPruner. We observe that when pruning at shallow layers, even with more tokens retained, performance still significantly decreases, indicating that attention distribution from shallow layers is not reliable, which aligns with the observations in Figure 2. Then the performance begins to stabilize from 12-th layer onwards, and we finally select $k$=12 which achieves the best performance.

**Ablation on retention ratio between $R_1$ and $R_2$.** The ratio between $R_1$ and $R_2$ determines the number of tokens retained at each stage. A smaller ratio leads to more visual information loss, while a larger ratio leads to a limited token budget allocated to text-related regions. As shown in Table 7b, the best result is achieved at $R_1 : R_2 = 3$, which is chosen in our default setting. It indicates that it is important to preserve the complete visual information in the shallow layers, while only a small number of tokens are required in the middle layers, which is consistent with our two-stage pruning strategy.

**Ablation on proportion of diverse tokens $\lambda$.** The hyper-parameter $\lambda$ controls the proportion of diverse tokens retained in the first stage. A low $\lambda$ risks losing comprehensive visual context, particularly background information that could also be crucial in cetrain scenarios. Conversely, a high $\lambda$ may allocate more tokens to less informative regions, leaving insufficient budget to adequately preserve salient foreground objects. As shown in Figure 5.(a), performance drops when $\lambda$ approaches

| k | $R_1$ | $R_2$ | RelAcc.(%) |
|---|---|---|---|
| 8 | 129 | 43 | 96.2 |
| 10 | 120 | 40 | 96.3 |
| 12 | 111 | 37 | **96.9** |
| 14 | 105 | 35 | 96.8 |
| 16 | 96 | 32 | **96.9** |

| $R_1 : R_2$ | $R_1$ | $R_2$ | RelAcc.(%) |
|---|---|---|---|
| 1 | 85 | 85 | 96.1 |
| 1.5 | 102 | 68 | 96.6 |
| 2 | 114 | 57 | 96.8 |
| 3 | 129 | 43 | **96.9** |
| 4 | 136 | 34 | 96.8 |

(a) Ablation study on pruning position $k$ in the LLM. $R_1$ and $R_2$ are the retained token numbers of the first and second pruning stages, respectively.

(b) Ablation study on retention ratio between $R_1$ and $R_2$.

Table 7: Ablation experiments for LearnPruner. We use LLaVA-v1.5-7B as the base model and evaluate the average performance on benchmarks mentioned in Table 1. Under default settings, pruning position $k$ is set to 12 and the number of retained tokens between two pruning stages $R_1 : R_2$ is fixed to 3.

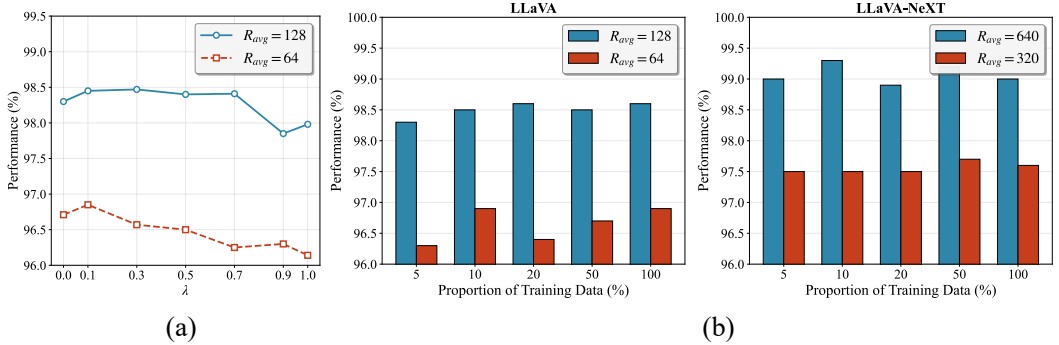

(a)  (b)

Figure 5: Ablation experiments for LearnPruner. (a). Ablation study on proportion of diverse tokens $\lambda$ in the first pruning stage. We use LLaVA-v1.5-7B as the base VLM. (b). Ablation study on different training data scales. We use LLaVA-v1.5-7B and LLaVA-NeXT-7B as the base VLM and train multiple models with different proportions (5%–100%) of the LLaVA-665K dataset.

1, as most VQA tasks focus on foreground objects. Finally, we empirically set $\lambda = 0.1$, which achieves the optimal balance between preserving semantic-rich foreground content and maintaining necessary background context across diverse visual scenarios.

**Data efficiency.** To evaluate the impact of training data scale on LearnPruner, we train multiple models with different proportions (5%–100%) of the LLaVA-665K dataset. As shown in Figure 5.(b), LearnPruner achieves strong performance with only 5% of the training data, and the model maintains consistent performance across different data ratios. This demonstrates the robustness and efficiency of LearnPruner in training, which can be attributed to the lightweight and concise design of our LPM module. Such data efficiency makes LearnPruner highly practical and flexible for deployment in real-world scenarios.

### A.3 QUALITATIVE RESULTS

To better understand the effectiveness of LearnPruner, we visualize the pruning results on the examples chosen from GQA and TextVQA in Figure 6 and Figure 7. We select FastV Chen et al. (2024) and VisionZip Yang et al. (2025) for comparison, which represent pruning approaches conducted after the vision encoder and within the LLM, respectively. VisionZip relies on `[CLS]` attention for pruning, which fails to properly focus on foreground objects and even selects meaningless padding tokens. On the other hand, limited by token budget, FastV prunes tokens after the second layer of the LLM, showing a bias toward preserving tokens in the lower half of the image and weak responsiveness to queries. In contrast, LearnPruner first removes visual redundancy after the vision encoder, accurately focusing on semantically rich regions through learning-based importance criterion, and

supplements background information with a small number of diversity tokens. In the second stage, leveraging reliable attention results from middle layers of the LLM, only the most query-relevant tokens are retained, enabling the VLM to answer questions using the minimal number of vision tokens. Futhermore, we show several failure cases in Figure 7. When dealing with complex visual scenes, the first-stage pruning may fail to adequately preserve critical visual information within the constrained token budget, leading to information loss that propagates through subsequent layers. Alternatively, even when query-relevant visual tokens are well retained, prediction failures can still occasionally occur, which is mainly due to the inherent reasoning limitations of the base VLM.

### A.4 EVALUATION BENCHMARKS

In this section, we provide a brief introduction about the evaluation benchmarks in our experiments.

**GQA Hudson & Manning (2019).** The GQA benchmark focuses on visual scene understanding and reasoning, leveraging scene graph structures from Visual Genome Krishna et al. (2017). It involves spatial relations and object attributes, making it challenging for models to achieve precise visual reasoning in complex scenes.

**ScienceQA Lu et al. (2022).** The ScienceQA benchmark uses multiple-choice questions to evaluate the zero-shot generalization on diverse science topics. The dataset contains rich domain diversity across three subjects: natural sciences,language science, and social science. Since some questions are not related to the image, we only evaluate the performance on the samples with images, denoted as "SQA$^I$" in the experimental tables.

**TextVaQA Singh et al. (2019).** The TextVQA benchmark evaluates model's ability to process and understand text information in images. Answers to the questions may be directly derived from the text in the images or require contextual reasoning. Moreover, optical character recognition (OCR) results is provided to assist the model to recognize text in the images. The dataset is denoted as "TextVQA$^T$" in the experimental tables.

**POPE Li et al. (2023b).** The POPE benchmark focuses on severe object hallucination issues in VLMs hence the questions mainly concern the presence of objects in the images. The reported result is calculated by the mean F1 score over the three indicators: adversarial, random, and popular.

**MME Fu et al. (2023).** The MME benchmark measures both perception and cognition abilities of VLMs on a total of 14 subtasks. Apart from OCR, the perception includes the recognition of coarse-grained and fine-grained objects. The former identifies the existence, count, position, and color of objects. The latter recognizes movie posters, celebrities, scenes, landmarks, and artworks. The cognition includes commonsense reasoning, numerical calculation, text translation, and code reasoning.

**VQA-v2 Goyal et al. (2017).** VQA-v2 is a large-scale benchmark consisting of 265,016 images from MSCOCO dataset Lin et al. (2014). Each image is paired with open-ended questions and 10 human-provided ground truth answers.

**MMBench Liu et al. (2024c).** MMBench is a bilingual benchmark for assessing the multi-modal capabilities of VLMs, which incorporates multiple-choice questions in both English and Chinese versions. It defines three levels of ability dimensions: Level-1 (Perception and Reasoning), Level-2 (six sub-abilities), and Level-3 (twenty specific tasks). We use 'MMB' and 'MMB$^{CN}$' to denote the English and Chinese versions of MMBench, respectively.

**TGIF-QA Jang et al. (2017).** The TGIF-QA benchmark extends the image-based VQA tasks to the video domain, requiring the model to understand and reason about spatial-temporal relationships in dynamic visual content. It includes 72K animated GIFs from the Tumblr GIF dataset Li et al. (2016) and 165K QA pairs. We employ GPT-4.1 to assist in evaluating the accuracy of the model's answers (same for the following two benchmarks).

**MSVD-QA Xu et al. (2017).** The MSVD-QA benchmark is constructed from the Microsoft Research Video Description Corpus Chen & Dolan (2011) and comprises 1,970 short video clips with 50,500 corresponding question-answer pairs.

**MSRVTT-QA Xu et al. (2017).** The MSRVTT-QA benchmark is based on the Microsoft Research Video to Text dataset Xu et al. (2016), it contains 10,000 video clips and 243,000 QA pairs. The

videos in MSRVTT-QA depict more complex scenes and activities than those in MSVD-QA, requiring models to effectively process both visual and temporal information.

### A.5 LLM USAGE DISCLOSURE STATEMENT

We utilized OpenAI's GPT-4 for assistance with language editing (improving grammar and clarity) and for generating/debugging experimental code and evaluating some benchmarks. All LLM-generated content was thoroughly reviewed, verified, and edited by the authors, who take full responsibility for the accuracy and integrity of this submission.

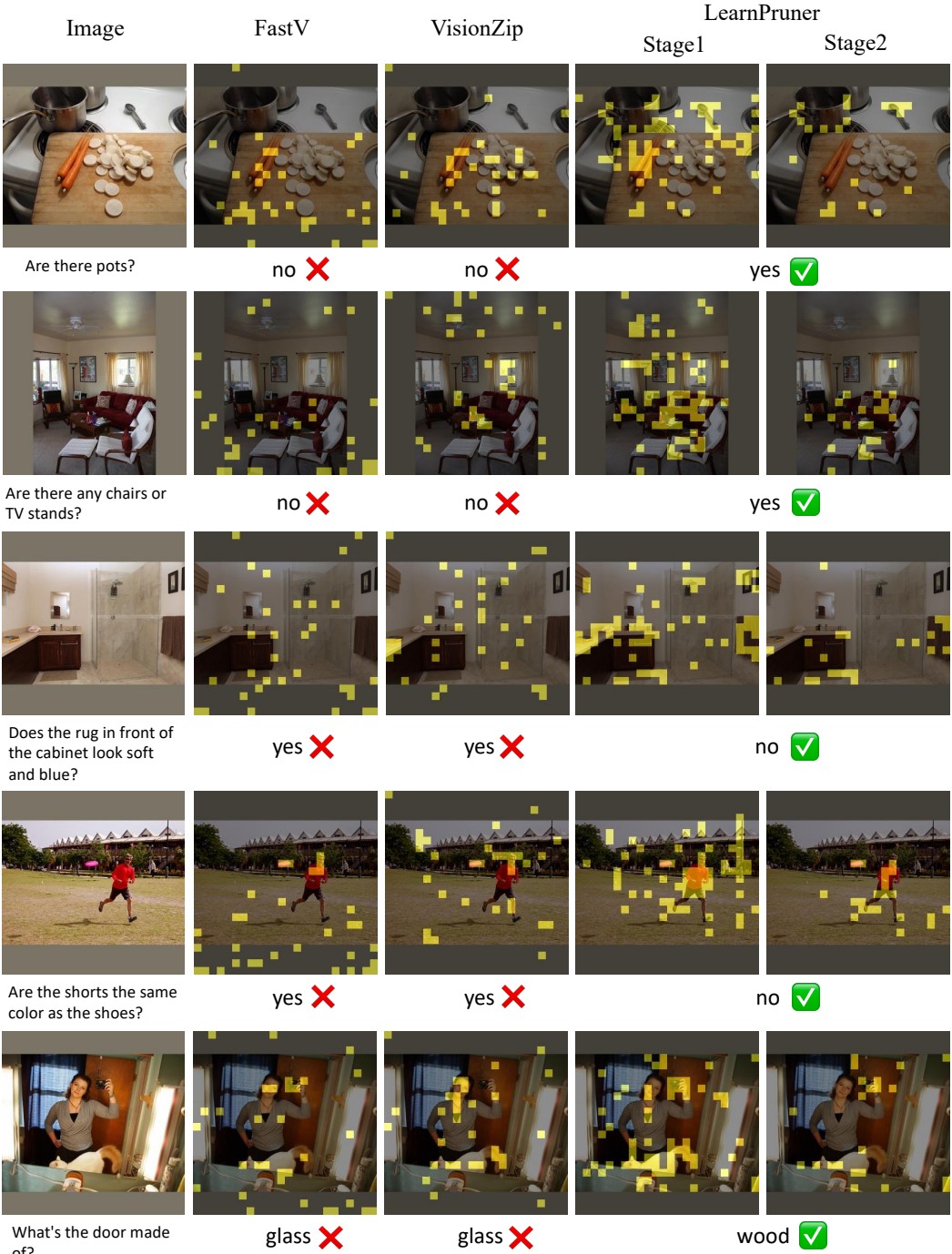

Figure 6: Visualization of the pruning results. Yellow patches indicate retained tokens, with an average of 32 tokens preserved.

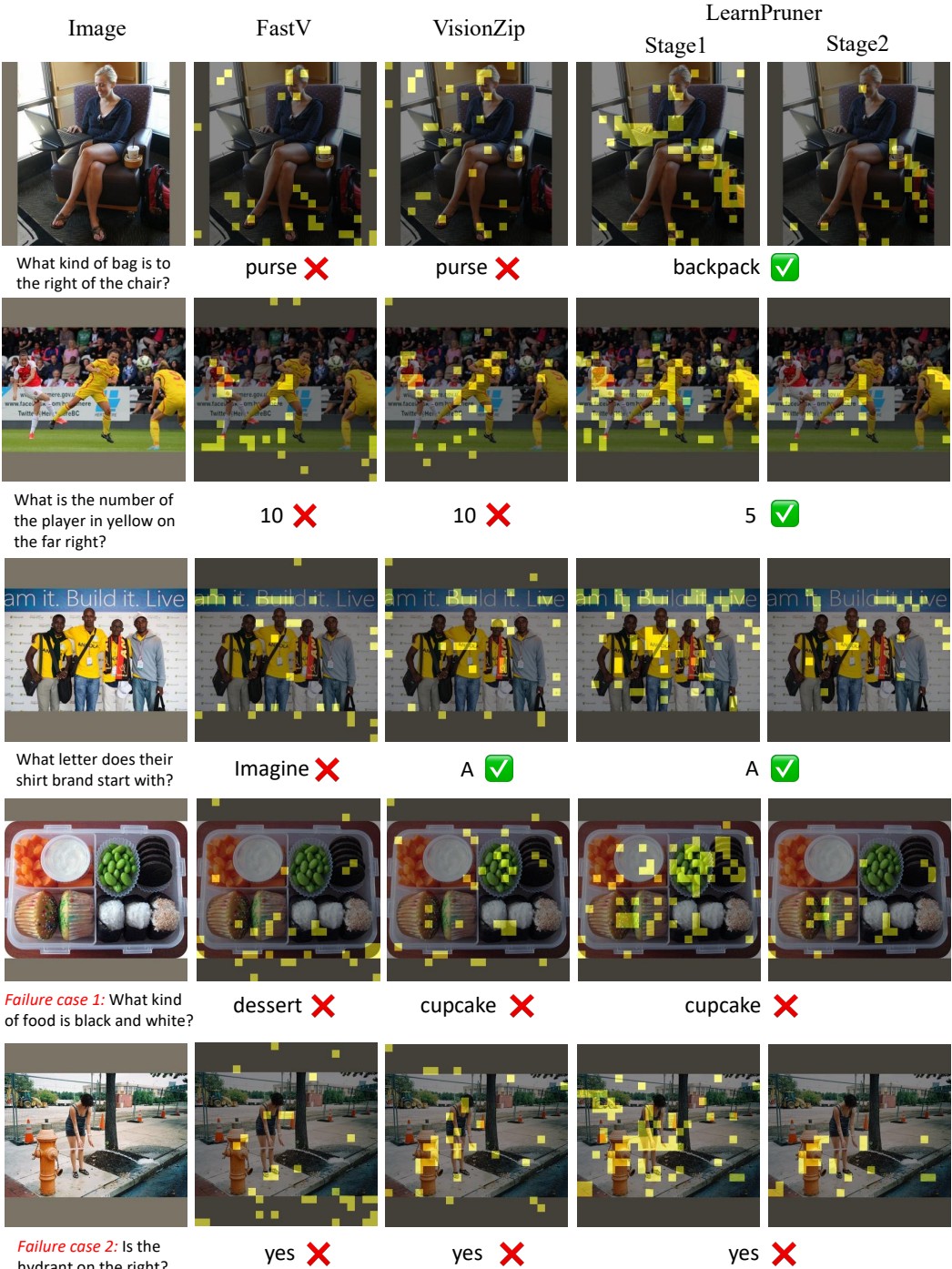

Figure 7: Visualization of the pruning results. Yellow patches indicate retained tokens, with an average of 32 tokens preserved.

