# OpenReview forum: "LearnPruner: Rethinking Attention-based Token Pruning in Vision Language Models"
_ICLR.cc/2026/Conference — ICLR 2026 Poster_

### Official Review · Reviewer_ui91 · 2025-10-25

**Soundness:** 2
**Presentation:** 3
**Contribution:** 2
**Rating:** 2
**Confidence:** 4

**Summary:**

This work presents a two-stage pruning framework designed to make VLM inference more efficient while preserving accuracy. The first stage employs a learnable module after the vision encoder to estimate and discard redundant visual tokens, and the second stage applies a text-guided pruning process in the LLM’s middle layers, selectively retaining only tokens that are relevant to the query through text-to-vision attention.

**Strengths:**

- This study offers an insightful examination of the attention behavior in both the vision encoder and the LLM, pointing out the shortcomings of [CLS]-based attention and demonstrating how LLM attention can be effectively exploited for token selection.
- Unlike prior works relying on attention heuristics ([CLS] or average attention), the learnable pruning module introduces a differentiable and data-driven importance predictor.
- The clear organization and presentation make the paper easy to follow.

**Weaknesses:**

- The concept of the learnable pruning module appears highly similar to the learnable image-token predictor proposed in Dynamic-LLaVA (ICLR 2025), which leverages image token features to predict a binary mask for selecting tokens during prefill computation. However, the current work does not include a proper comparison or discussion of this prior work. It would considerably strengthen the paper to include Dynamic-LLaVA as a baseline in the experiments and to explicitly discuss methodological differences and advantages, thereby clarifying the novelty and demonstrating the superiority of the proposed approach.
  * https://openreview.net/forum?id=hzVpZDrW73

- The paper briefly mentions a diversity-based token selection module to preserve background information during inference. However, the description of this component is insufficient and does not clearly explain how the module is implemented. Moreover, I was not able to find an ablation study quantifying its contribution, making it difficult to assess the actual impact of the diversity-based selection compared to the LPM.

- Diversity-based vision token pruning methods are not properly discussed or compared. It would strengthen the paper to include these approaches as baselines in the comparison experiments to better demonstrate the superiority of the proposed method.
  * DivPrune: Diversity-based Visual Token Pruning for Large Multimodal Models ( https://arxiv.org/abs/2503.02175 )
  * DART: Stop Looking for Important Tokens in Multimodal Language Models: Duplication Matters More ( https://arxiv.org/abs/2502.11494 )

- I am concerned about the practical applicability of the proposed method in conjunction with modern high-performance inference engines and optimization techniques. The use of middle-layer text-to-vision attention in the second stage may not be directly compatible with implementations such as FlashAttention. Furthermore, introducing additional parameters for the LPM in the first stage and performing token pruning within intermediate layers in the second stage could require substantial implementation effort and modification for popular inference frameworks such as vLLM or TensorRT. It would be beneficial for the authors to discuss these integration challenges and clarify how their method can be efficiently deployed in real-world inference systems.

**Questions:**

Please refer to the Weaknesses section.

---

> ### Author Response · Authors · 2025-11-25
> **Response to Reviewer ui91 (Part 1 / 2)**
>
> We sincerely thank Reviewer ui91 for the valuable and insightful feedback. Our responses according to the reviewer's comments are summarized as follows.
>
> > The concept of the learnable pruning module appears highly similar to the learnable image-token predictor proposed in Dynamic-LLaVA (ICLR 2025), which leverages image token features to predict a binary mask for selecting tokens during prefill computation. However, the current work does not include a proper comparison or discussion of this prior work. It would considerably strengthen the paper to include Dynamic-LLaVA as a baseline in the experiments and to explicitly discuss methodological differences and advantages, thereby clarifying the novelty and demonstrating the superiority of the proposed approach.
>
> Thank you for raising this point about Dynamic-LLaVA. We acknowledge the similarity in module design between our LPM and Dynamic-LLaVA's learnable image-token predictor. We did not perform a direct performance comparison with Dynamic-LLaVA because it requires fine-tuning the entire base VLM, whereas our approach only trains the lightweight LPM, making a fair comparison difficult.
>
> That said, we report the performance of single-stage pruning (using only LPM, similar conceptually to Dynamic-LLaVA's approach) in our ablation study Table 6, which shows that relying solely on LPM achieves lower performance than our two-stage approach, highlighting that our key innovation lies in adopting appropriate pruning criteria in different stages.
>
> Specifically, based on our analysis in Section 3.2, in early stages (vision encoder and shallow LLM layers), attention signals are unreliable. Therefore, we employ a learning-based approach (LPM) to remove inherent visual redundancy. In deeper layers where attention becomes reliable, we leverage text-to-vision attention to further remove query-irrelevant content. This differentiated strategy enables more effective budget allocation.
>
> In the revised manuscript, we have added explicit discussion of Dynamic-LLaVA in the ablation study section to clarify the methodological differences.
>
> > The paper briefly mentions a diversity-based token selection module to preserve background information during inference. However, the description of this component is insufficient and does not clearly explain how the module is implemented. Moreover, I was not able to find an ablation study quantifying its contribution, making it difficult to assess the actual impact of the diversity-based selection compared to the LPM.
>
> The diversity-based token selection module operates as follows:
> 1. Start with the token set selected by LPM based on predicted importance scores (e.g., 90% of the budget)
> 2. For each remaining token, compute its cosine similarity with all currently selected tokens
> 3. Calculate the maximum similarity for each candidate token
> 4. Add the token with the smallest maximum similarity to the selected set
> 5. Repeat steps 2-4 until the total token budget is reached
>
> We have refined the description in Section 3.3 to better explain the algorithm, we hope this addresses your concern and clarifies the implementation.
>
> Moreover, we have included a new ablation study on the diversity-based token selection component in the revised manuscript in Appendix A.2.  The hyper-parameter $\lambda$ is used to control the proportion of diverse tokens retained in the first stage. A low $\lambda$ risks losing comprehensive visual context, particularly background information that could also be crucial in cetrain scenarios. Conversely, a high $\lambda$ may allocate more tokens to less informative regions, leaving insufficient budget to adequately preserve salient foreground objects. As shown in Figure 5(a), performance drops when $\lambda$ approaches 1, as most VQA tasks focus on foreground objects. Finally, we empirically set $\lambda=0.1$, which achieves the optimal balance between preserving semantic-rich foreground content and maintaining necessary background context across diverse visual scenarios.
>
>
> ### Ablation study on proportion of diverse tokens $\lambda$ in the first pruning stage
>
> | $\lambda$ | $R_{avg} = 128$ | $R_{avg} = 64$ |
> |:---------:|:---------------:|:--------------:|
> | 0.0       | 98.30           | 96.71          |
> | 0.1       | 98.45           | 96.85          |
> | 0.3       | 98.47           | 96.57          |
> | 0.5       | 98.40           | 96.50          |
> | 0.7       | 98.41           | 96.25          |
> | 0.9       | 97.85           | 96.30          |
> | 1.0       | 97.98           | 96.14          |

---

> ### Author Response · Authors · 2025-11-25
> **Response to Reviewer ui91 (Part 2 / 2)**
>
> > Diversity-based vision token pruning methods are not properly discussed or compared. It would strengthen the paper to include these approaches as baselines in the comparison experiments to better demonstrate the superiority of the proposed method.
>
> Thank you for this constructive feedback. We have added description of diversity-based pruning methods (DivPrune and DART) in the Related Work (Section 2.2) and included experimental comparisons with these methods in Table 1 and Table 2. The results demonstrate that LearnPruner consistently outperforms these diversity-based approaches. We appreciate this suggestion as it helps provide a more complete comparison.
>
> > I am concerned about the practical applicability of the proposed method in conjunction with modern high-performance inference engines and optimization techniques. The use of middle-layer text-to-vision attention in the second stage may not be directly compatible with implementations such as FlashAttention. Furthermore, introducing additional parameters for the LPM in the first stage and performing token pruning within intermediate layers in the second stage could require substantial implementation effort and modification for popular inference frameworks such as vLLM or TensorRT. It would be beneficial for the authors to discuss these integration challenges and clarify how their method can be efficiently deployed in real-world inference systems.
>
> Thank you for raising this important practical concern. We would like to clarify the compatibility and implementation feasibility of our method with modern inference systems.
>
> **FlashAttention Compatibility**:
>
> For layers that do not require attention scores, FlashAttention can be used normally without modification. For the specific layer where we need attention scores, we can adopt the dual-flash attention proposed by SparseVLM:  the first pass operates identically to the original FlashAttention, and the second pass uses a specially designed value matrix to directly extract average attention scores. Since this operation is only required once at the k-th layer during the prefill phase, it introduces minimal overhead while maintaining compatibility with FlashAttention.
>
> **Integration with Inference Frameworks (vLLM, TensorRT)**:
>
> *First Stage*: Token pruning occurs before entering the LLM, requiring no modifications to the LLM inference pipeline.
>
> *Second Stage*: The pre-defined pruning ratio allows us to pre-calculate token counts and allocate memory appropriately. At the k-th layer during prefill, we perform top-k selection of vision tokens based on attention scores and concatenate the retained tokens. All subsequent operations, including paging and memory management, proceed identically to the base model. This localized modification at a single layer does not require fundamental changes to the inference framework, making the implementation straightforward and practical.

---

> ### Comment · Reviewer_ui91 · 2025-11-27
>
> I appreciate the authors' time and effort for this rebuttal. Most concerns are resolved with clarification and additional experiments, and therefore I will increase my score from 2 to 4.
>
> The comparison with the concept of Dynamic-LLaVA based on Table 6 is understandable, but it would be more convincing to include a direct empirical comparison. According to Table 1 (LLaVA-1.5-7B) in the Dynamic-LLaVA paper, Dynamic-LLaVA with 80% token pruning achieves 78.0-77.9% accuracy on VQAv2, whereas LearnPruner with a similar pruning ratio (77.8%) reports 77.3%. Likewise, Dynamic-LLaVA reports 61.4-61.3% accuracy on GQA, while LearnPruner shows 60.3%. Given these numbers, I am not fully convinced about the superiority of LearnPruner over Dynamic-LLaVA. Please correct me if I am wrong.
>
> Regarding the comments on compatibility with modern inference systems, while this does not influence my score, it would be valuable to include actual implementation results. For FlashAttention compatibility, quantifying the overhead in time and memory introduced by the one-time operation at the k-th layer during the prefill phase would make the claim more concrete. In addition, based on my experience, modifying vLLM is not entirely straightforward (though certainly possible), so validating the method through a working vLLM integration would significantly strengthen the practical impact.

---

> > ### Author Response · Authors · 2025-12-03
> > **Response to Reviewer ui91**
> >
> > Thank you very much for your time and constructive feedback. We greatly appreciate your recognition of our efforts and are grateful for the score increase. Our responses according to the reviewer's comments are summarized as follows.
> >
> > > The comparison with the concept of Dynamic-LLaVA based on Table 6 is understandable, but it would be more convincing to include a direct empirical comparison.
> >
> > To further address your concern about the comparison with Dynamic-LLaVA, we have conducted additional experiments following Dynamic-LLaVA's training and testing settings for a fair comparison:
> >
> > **Training Setup**: We train on the full LLaVA-665K dataset and fine-tune all LLM parameters (rather than only training the LPM)
> >
> > **Testing Setup**: Dynamic-LLaVA retains all 576 tokens in the first two LLM layers and only 115 tokens from the third layer onward. To align with this, we set the average number of retained tokens to 144 (calculated as $\frac{576 \times 2 + 115 \times 30}{32} = 144$).
> >
> > **Results**: As shown in Table below,  LearnPruner outperforms Dynamic-LLaVA by 1.5\% on average across benchmarks. Remarkably, using only 25\% of vision tokens , our method achieves even better performance than the base VLM, demonstrating the superiority of our method.
> >
> > ### Comparison with Dynamic-LLaVA
> >
> > | Model | Token | GQA | SQA-IMG | TextVQA | POPE | MME | VQAv2 | MMB | Avg |
> > | :--- | :---: | :---: | :---: | :---: | :---: | :---: | :---: | :---: | :---: |
> > | **llava-v1.5-7b** | 576 | 61.9 | 69.5 | 58.2 | 85.9 | 1510.7 | 78.5 | 64.7 | 100% |
> > | **dynamic-llava** | 144 | 61.4 | 69.1 | 57.0 | 85.0 | 1479.8 | 78.0 | 65.4 | 99.13% |
> > | | | *99.19%* | *99.42%* | *97.94%* | *98.95%* | *97.95%* | *99.36%* | *101.08%* | |
> > | **learnpruner** | 144 | 62.4 | 69.8 | 58.3 | 86.7 | 1512.3 | 78.9 | 65.6 | 100.64% |
> > | | | *100.79%* | *100.45%* | *100.22%* | *100.97%* | *100.10%* | *100.56%* | *101.39%* | |
> >
> > >  For FlashAttention compatibility, quantifying the overhead in time and memory introduced by the one-time operation at the k-th layer during the prefill phase would make the claim more concrete.
> >
> > Thank you for the valuable suggestion regarding practical implementation validation. We have conducted additional experiments to quantify the overhead of attention extraction at the k-th layer during the prefill phase.
> > We evaluated three implementation approaches:
> > 1. **FlashAttention Upper Bound**: Standard FlashAttention without attention extraction (ideal case, for reference)
> > 2. **Eager Attention**: Using standard attention implementation (eager) at the k-th layer to extract attention scores
> > 3. **Dual FlashAttention**: Executing FlashAttention twice at the k-th layer, as described in our previous response
> >
> > We use LLaVA-v1.5-7B and LLaVA-NeXT-7B to perform the inference on POPE dataset. As shown in Table  below, the overhead varies with sequence length. For LLaVA-v1.5, which has relatively short token sequences, both Eager Attention and Dual FlashAttention add negligible latency. However, for LLaVA-NeXT with longer sequences (average 640 tokens retained), Eager Attention incurs noticeable inference delay, while Dual FlashAttention introduces only minimal overhead, as it requires only one additional FlashAttention computation at the k-th layer.
> > These results confirm that LearnPruner is both feasible and effective for practical deployment, achieving significant inference acceleration while maintaining compatibility with modern attention mechanisms.
> >
> > ### Efficiency Analysis on LLaVA-v1.5
> >
> > | Method | Implementation | Token | Prefill Time (s) | Total Time (s) |
> > | :--- | :--- | :---: | :---: | :---: |
> > | **llava-v1.5** | FlashAttention | 576 | 544.47 | 874.13 |
> > | **learnpruner** | FlashAttention Upper Bound | 128 | 293.98 | 708.05 |
> > | | | 32 | 228.34 | 586.44 |
> > | | Eager Attention | 128 | 296.4 | 709.15 |
> > | | | 32 | 233.65 | 590.05 |
> > | | Dual FlashAttention | 128 | 297.59 | 711.59 |
> > | | | 32 | 234.05 | 590.18 |
> >
> > ### Efficiency Analysis on LLaVA-NeXT
> >
> > | Method | Implementation | Token | Prefill Time (s) | Total Time (s) |
> > | :--- | :--- | :---: | :---: | :---: |
> > | **llava-next** | FlashAttention | 2880 | 1598.41 | 1987.33 |
> > | **learnpruner** | FlashAttention Upper Bound | 640 | 583.64 | 972.61 |
> > | | | 160 | 312.53 | 697.83 |
> > | | Eager Attention | 640 | 620.18 | 1021.16 |
> > | | | 160 | 321.38 | 718.11 |
> > | | Dual FlashAttention | 640 | 588.86 | 981.36 |
> > | | | 160 | 318.88 | 709.05 |

---

### Official Review · Reviewer_FyhA · 2025-10-31

**Soundness:** 3
**Presentation:** 2
**Contribution:** 3
**Rating:** 6
**Confidence:** 2

**Summary:**

This work proposes LearnPruner, a two-stage pruning approach for VLMs. The proposed approach first removes redundant vision tokens with a learnable pruning module after the vision encoder, and then retains only task-relevant tokens in the LLM's middle layer. These are based on some analyses on the VLM's attention patterns: in visual encoder, [CLS] fails to adequately attend to salient foreground objects, while in LLM, text-to-vision attention can provide reliable guidance for token selection. Experiments with multiple models and a variety of datasets show the effectiveness of the proposed approach.

**Strengths:**

- The direction of making VLMs more efficiency is important for its real-world deployment.
- The method is well-motivated by preliminary analyses.
- The proposed approach is shown to obtain good performance.

**Weaknesses:**

- It is unclear how well the Learnable Pruning Module can generalize to different tasks, that may be very different to the ones that have been seen in the training of the module.
- The ablation study seems a little thin, where it would be better if more settings can be investigated. For example, what will be a good ratio of the compression of the vision encoder and the LLM? (The ratio is set to 3, but how would the overall efficiency be influenced if we have other settings?)

**Questions:**

- I'm wondering how some of the hyper-parameters (such as k=12-th layer of the LLM for the second stage) are selected? Will these settings be model- or benchmark-sensitive?
- The approach seems to be query sensitive, if we would like to perform pruning in a query-agnostic way, will the proposed approach be adaptable to this setting?

---

> ### Author Response · Authors · 2025-11-25
> **Response to Reviewer FyhA (Part 1 / 2)**
>
> We sincerely thank Reviewer FyhA for the valuable feedback. We have provided the following responses to address the Weaknesses and Questions you raised, and we hope these clarifications resolve your concerns.
>
> > It is unclear how well the Learnable Pruning Module can generalize to different tasks, that may be very different to the ones that have been seen in the training of the module.
>
> For further comprehensive evaluation, we added evaluation on SeedBench and MMMU. As shown below, LearnPruner maintains strong performance on both benchmarks. Notably, MMMU contains college-level questions across diverse specialized domains with heterogeneous image types (charts, diagrams, chemical structures)—content substantially different from the general VQA tasks in our training data (LLaVA-665K), demonstrating strong out-of-distribution generalization.
>
> Besides, our Video-LLaVA experiments show that despite training only on image data, the LPM generalizes well to video tasks, providing another perspective on generalization.
>
> ### More benchmark results on LLaVA-v1.5-7B
>
> | Dataset    | LLaVA-v1.5-7B | 128 tokens        |                  | 64 tokens         |                   |
> |------------|---------------|-------------------|-------------------|-------------------|-------------------|
> |            |               | FastV             | LearnPruner       | FastV             | LearnPruner       |
> | SeedBench  | 66.2          | 56.1              | 64.5              | 41.7              | 62.7              |
> |            | 100%          | 84.8%             | 97.5%             | 63.1%             | 94.7%             |
> | MMMU       | 32.0          | 31.3              | 31.6              | 30.3              | 31.5              |
> |            | 100%          | 97.8%              | 98.8%             | 94.7%             | 98.5%             |
>
> >The ablation study seems a little thin, where it would be better if more settings can be investigated. For example, what will be a good ratio of the compression of the vision encoder and the LLM? (The ratio is set to 3, but how would the overall efficiency be influenced if we have other settings?)
>
> Thank you for this feedback. We would like to point out that ablation studies on the R1:R2 compression ratio were already included in Appendix A.3.2 of the original submission, where we examined different ratios (1, 1.5, 2, 3, 4) and their impact on model performance. As shown in Table.7(b), the best result is achieved at $R_1:R_2=3$, which is chosen in our default setting.
> It indicates that it is important to preserve the complete visual information in the shallow layers, while only a small number of tokens are required in the middle layers, which is consistent with our two-stage pruning strategy.
>
> Moreover, we appreciate your suggestion for more comprehensive ablation studies. In the revised manuscript, we have added additional ablation study on diversity-based token selection and training data sclaes in Appendix A.2. We believe these experiments provide a comprehensive understanding of our method's design choices and their effects on performance.

---

> ### Author Response · Authors · 2025-11-25
> **Response to Reviewer FyhA (Part 2 / 2)**
>
> > I'm wondering how some of the hyper-parameters (such as k=12-th layer of the LLM for the second stage) are selected? Will these settings be model- or benchmark-sensitive?
>
> Thank you for this question. We would like to clarify that the ablation study on the pruning layer $k$ was already included in the Appendix of the original submission.
>
> The selection of $k=12$ is guided by our analysis in Figure 2(c), which shows that text-to-vision attention becomes reliable and stable starting from the 12-th layer of the LLM. The ablation study in Appendix X further validates this choice by evaluating different values of $k$ (8, 10, 12, 14, 16), demonstrating that k=$12$ achieves optimal performance.
>
> Regarding model and benchmark sensitivity, we provide more detailed experimental results below showing LearnPruner's performance on different base VLMs (LLaVA-v1.5 and Qwen2.5-VL) across multiple benchmarks. The results show that LearnPruner maintains consistent performance across different model architectures and benchmark tasks.
>
> ### Ablation study on $k$ for LLaVA-v1.5-7B
>
> | Model         | GQA   | SQA-IMG | TextVQA | POPE  | MME     | Avg     |
> |---------------|-------|---------|---------|-------|---------|---------|
> | LLaVA-v1.5-7b   | 61.9  | 69.5    | 58.2    | 85.9  | 1862    | 100%       |
> | **Retain 64 tokens** |       |         |         |       |         |         |
> | k=8           | 58.33 | 67.77   | 55.80   | 84.64 | 1772.39 | 96.27%  |
> |               | 94.23%| 97.51%  | 95.88%  | 98.53%| 95.19%  |         |
> | k=10          | 58.34 | 67.77   | 56.13   | 84.69 | 1778.99 | 96.47%  |
> |               | 94.25%| 97.51%  | 96.44%  | 98.59%| 95.54%  |         |
> | k=12          | 58.9  | 68.3    | 56.6    | 86.8  | 1750.0  | 97.14%  |
> |               | 95.15%| 98.27%  | 97.25%  | 101.05%| 93.98% |         |
> | k=14          | 58.89 | 68.00   | 56.56   | 85.92 | 1752.99 | 96.87%  |
> |               | 95.14%| 97.84%  | 97.18%  | 100.02%| 94.15% |         |
> | k=16          | 58.70 | 68.07   | 56.36   | 86.45 | 1767.48 | 97.04%  |
> |               | 94.83%| 97.94%  | 96.84%  | 100.64%| 94.92% |         |
>
> ### Ablation study on $k$ for Qwen2.5-VL-7B
>
> | Model         | GQA   | SQA-IMG | TextVQA | MME      | Avg     |
> |---------------|-------|---------|---------|----------|---------|
> | Qwen2.5-VL-7B     | 59.67 | 77.39   | 76.67   | 2324.36  | 100%       |
> | **Retain 320 tokens** |       |         |         |          |         |
> | k=8           | 59.5  | 71.2    | 74.9    | 2361.3   | 97.73%  |
> |               | 99.7% | 91.9%   | 97.7%   | 101.6%   |         |
> | k=10          | 57.9  | 77.2    | 73.7    | 2356.3   | 98.59%  |
> |               | 97.0% | 99.8%   | 96.1%   | 101.4%   |         |
> | k=12          | 59.0  | 77.0    | 74.4    | 2357.5   | 99.16%  |
> |               | 98.8% | 99.4%   | 97.0%   | 101.4%   |         |
> | k=14          | 58.0  | 74.5    | 74.2    | 2341.0   | 97.74%  |
> |               | 97.2% | 96.3%   | 96.8%   | 100.7%   |         |
> | k=16          | 58.8  | 77.6    | 74.7    | 2337.7   | 99.22%  |
> |               | 98.5% | 100.3%  | 97.4%   | 100.6%   |         |
>
> > The approach seems to be query sensitive. If we would like to perform pruning in a query-agnostic way, will the proposed approach be adaptable to this setting?
>
> Thank you for this question. The first pruning stage (after the vision encoder) in LearnPruner is inherently query-agnostic, it removes visual redundancy based solely on image content without requiring any textual input. For query-agnostic scenarios, LearnPruner can operate using only the first stage. As demonstrated in Table 6, this approach outperforms common query-agnostic methods such as [CLS] attention-based pruning.

---

### Official Review · Reviewer_uDiD · 2025-10-31

**Soundness:** 2
**Presentation:** 2
**Contribution:** 2
**Rating:** 4
**Confidence:** 4

**Summary:**

LearnPruner is a two-stage token pruning framework for Vision-Language Models. It first uses a learnable module to remove redundant vision tokens, then prunes text-irrelevant tokens in LLM middle layers. It preserves 95% performance with only 5.5% tokens, achieving 3.2× faster inference and superior accuracy-efficiency trade-off.

**Strengths:**

1. The writing is clear and easy to follow.
2. Experimental results show strong performance on both LLaVA-1.5 and LLaVA-Next.

**Weaknesses:**

1. Experiments on LLaVA-1.5 and LLaVA-Next provide limited persuasiveness; please include comparisons on Qwen2.5-VL.
2. The design of Remove Text-Irrelevant Content does not appear novel, it seems to be a straightforward reuse of existing ideas.
3. Additional training is required compared with training-free methods.

**Questions:**

see weaknesses

---

> ### Author Response · Authors · 2025-11-25
> **Response to Reviewer uDiD**
>
> We sincerely thank the reviewer uDiD for the efforts in reviewing our paper. Our responses according to the reviewer's comments are summarized as follows.
>
> > Experiments on LLaVA-1.5 and LLaVA-Next provide limited persuasiveness; please include comparisons on Qwen2.5-VL.
>
> Regarding your concern about generalizability and the request for comparisons on Qwen2.5-VL, we would like to clarify that these results were included in our original submission, located in Appendix A.3.1 (Table 6). To highlight the importance of this experiment, in our revised manuscript, we have now moved this section to the main paper (Section 4.2, Table 4) to improve visibility and address your concern directly.
>
> > The design of Remove Text-Irrelevant Content does not appear novel, it seems to be a straightforward reuse of existing ideas.
>
> Thank you for raising this point. We acknowledge that using attention in LLM as the pruning metric is not novel per se. However, our motivation stems from the attention shift problem proposed in VisPruner, which questions the reliability of LLM attention for pruning. Through both quantitative and qualitative analysis in Section 3.2, we demonstrate that text-to-vision attention becomes reliable and stable starting from the middle layers of the LLM, which provides the foundation for our second pruning stage.
>
> Moreover, our key innovation lies in adopting appropriate pruning criteria in different stages. Prior multi-stage pruning methods, such as SparseVLM and PyramidDrop, typically rely on attention-based metrics across all stages. However, as our experiments reveal, the effectiveness of attention varies significantly across different layers. In contrast, we first employ a learning-based metric to remove inherent visual redundancy in the first stage, and then leverage reliable attention patterns in the middle layers of the LLM to remove text-irrelevant content in the second stage, enabling more effective pruning.
>
> In summary, our contribution is not the individual components themselves, but rather the principled design of using the right metric at the right stage, guided by our comprehensive analysis of attention in vision encoder and LLM.
>
> >Additional training is required compared with training-free methods.
>
> We acknowledge that LearnPruner requires training compared to training-free methods, but this modest training cost brings substantial performance gains. As shown in Table 1 and Table 2, training-based approaches consistently outperform training-free approaches across different base VLMs and benchmarks. This superiority has motivated recent research toward learnable pruning strategies, as evidenced by concurrent works such as TwigVLM and ATP-LLaVA.
>
> Compared to these training-based approaches, LearnPruner achieves superior performance with significantly fewer trainable parameters (0.53M vs. 610M for TwigVLM). Moreover, as analyzed in the  Appendix A.2, the lightweight design of our LPM module also enables remarkable data efficiency, requiring only 5% of training data to achieve strong performance, which makes LearnPruner highly practical for real-world deployment.
>
> In summary, we believe the limited training overhead is acceptable for the performance gains and practicality of LearnPruner.

---

### Official Review · Reviewer_iSWN · 2025-11-01

**Soundness:** 3
**Presentation:** 3
**Contribution:** 3
**Rating:** 6
**Confidence:** 3

**Summary:**

This paper proposed a new token pruning pipeline, LearnPruner, with a learnable module to determine token importance. Despite the prior concern with the bias in vision-to-vision attention, the proposed method leverage the more robust text-to-vision attention for the second stage token pruning. In the experiments, the proposed pruning framework is able to better preserve accuracy compared many prior baselines while providing similar benefit in computational cost reduction.

**Strengths:**

- The learnable module provides a more robust way to acquire the visual token importance.
- The overall performance is improved compared to prior baselines.

**Weaknesses:**

- There is no failure case analysis.
- Ablation study on the LearnPruner training: what will happen if more or fewer training data samples are used?
- Extra ablation study on the diversity-based token selection is needed.
- The use of text-to-vision attention remain a bottleneck in the pruning process. The challenge of attention bias and drifting is not tackled with.

**Questions:**

See weaknesses.

---

> ### Author Response · Authors · 2025-11-25
> **Response to Reviewer iSWN (Part 1 / 2)**
>
> We sincerely thank the reviewer iSWN for the efforts in reviewing our paper. Our responses according to the reviewer's comments are summarized as follows.
> > There is no failure case analysis.
>
> Thank you for pointing out the necessity of the failure case analysis. In our revised manuscript, we have provided several failure cases in Figure 7  and a detailed discussion in Appendix A.3. The analysis reveals two primary sources of error:
> - When dealing with complex visual scenes, the first-stage pruning may fail to adequately preserve critical visual information within the constrained token budget, leading to information loss that propagates through subsequent layers.
> - Prediction failures can occasionally occur even when query-relevant visual tokens are well retained in both pruning stages. This is mainly attributable to the inherent reasoning limitations of the base VLM.
>
> > Ablation study on the LearnPruner training: what will happen if more or fewer training data samples are used?
>
> We have included a new ablation experiment in the revised manuscript to demonstrate data efficiency of our LearnPruner in Appendix A.2. We choose LLaVA-v1.5-7b and LLaVA-NeXT as the base VLM and train multiple models with different proportions (5\%--100\%) of the LLaVA-665K dataset. As shown in Figure.5(b) , LearnPruner achieves strong performance with only 5\% of the training data, and the model maintains consistent performance across different data ratios. This demonstrates the robustness and efficiency of LearnPruner in training, which can be attributed to the lightweight and concise design of our LPM module. Such data efficiency makes LearnPruner highly practical and flexible for deployment in real-world scenarios.
>
> ### Data efficiency for LLaVA-7B
>
> | Proportion of Training Data (%) | $R_{avg}= 128$ | $R_{avg}= 64$ |
> |---------------------------------|---------------------|--------------------|
> | 5                               | 98.3                | 96.3               |
> | 10                              | 98.5                | 96.9               |
> | 20                              | 98.6                | 96.4               |
> | 50                              | 98.5                | 96.7               |
> | 100                             | 98.6                | 96.9               |
>
> ### Data efficiency for LLaVA-NeXT-7B
>
> | Proportion of Training Data (%) | $R_{avg}=640$ | $R_{avg}=320$ |
> |---------------------------------|---------|---------|
> | 5                               | 99.0    | 97.5    |
> | 10                              | 99.3    | 97.5    |
> | 20                              | 98.9    | 97.5    |
> | 50                              | 99.2    | 97.7    |
> | 100                             | 99.0    | 97.6    |

---

> ### Author Response · Authors · 2025-11-25
> **Response to Reviewer iSWN (Part 2 / 2)**
>
> > Extra ablation study on the diversity-based token selection is needed.
>
> We have included a new ablation study on the diversity-based token selection component in the revised manuscript in Appendix A.2.  The hyper-parameter $\lambda$ is used to control the proportion of diverse tokens retained in the first stage. A low $\lambda$ risks losing comprehensive visual context, particularly background information that could also be crucial in cetrain scenarios. Conversely, a high $\lambda$ may allocate more tokens to less informative regions, leaving insufficient budget to adequately preserve salient foreground objects. As shown in Figure.5(a), performance drops when $\lambda$ approaches 1, as most VQA tasks focus on foreground objects. Finally, we empirically set $\lambda=0.1$, which achieves the optimal balance between preserving semantic-rich foreground content and maintaining necessary background context across diverse visual scenarios.
>
> ### Ablation study on proportion of diverse tokens $\lambda$ in the first pruning stage
>
> | $\lambda$ | $R_{avg} = 128$ | $R_{avg} = 64$ |
> |:---------:|:---------------:|:--------------:|
> | 0.0       | 98.30           | 96.71          |
> | 0.1       | 98.45           | 96.85          |
> | 0.3       | 98.47           | 96.57          |
> | 0.5       | 98.40           | 96.50          |
> | 0.7       | 98.41           | 96.25          |
> | 0.9       | 97.85           | 96.30          |
> | 1.0       | 97.98           | 96.14          |
>
> > The use of text-to-vision attention remain a bottleneck in the pruning process. The challenge of attention bias and drifting is not tackled with.
>
> In Figure.2(b), we have visualized the attention distributions in the VLM, revealing two key findings:
> - Attention bias within the vision modality is much more severe than text-to-vision attention.
> - While text-to-vision attention exhibits bias in averaged results, individual samples show high variance, with query-relevant regions maintaining strong attention signals that are not obscured by the bias.
>
> Moreover, through both quantitative and qualitative analysis in Section 3.2, we demonstrate that text-to-vision attention becomes reliable and stable starting from the middle layers of the LLM.
> Overall, our analysis shows that attention bias does not pose a significant bottleneck when the attention result is used in a proper way. Fully resolving this issue may require modifications to training paradigms or positional encoding.

---

### Author Response · Authors · 2025-12-03
**Summary of Response**

We would like to express our sincere gratitude to the Reviewers, Area Chairs, and Program Chairs for their time and dedicated effort. We have carefully studied all comments with detailed responses and additional experiments.

Our approach, LearnPruner, has been recognized for its effectiveness and clarity:
- Novelty and Insight: provides a more robust way to acquire the visual token importance (R_iSWN); insightful examination of the attention behavior in both the vision encoder and the LLM (R_ui91); well-motivated by preliminary analyses (R_FyhA)
- Performance: The proposed approach is shown to obtain good performance. (R_iSWN, R_uDiD and R_FyhA)
- Presentation: Clear organization and presentation make the paper easy to follow. (R_uDiD and R_ui91)

In response to the reviewers' comments, we have clarified key issues and incorporated revisions (highlighted in blue in the manuscript):

- Clarification of the novelty (R_uDiD, R_ui91). We clarified that our key innovation lies in adopting appropriate pruning criteria in different stages. Through comprehensive attention analysis, we demonstrate that attention reliability varies across layers (Sec. 3.2), leading to our principled design: learning-based pruning where attention is unreliable (after vision encoder), and attention-based pruning where it becomes reliable (middle LLM layers). This differentiated strategy enables superior pruning performance and effective budget allocation.
- Comparison with Dynamic-LLaVA and diversity-based pruning methods (R_ui91). For Dynamic-LLaVA (a learning-based efficient VLM approach conceptually similar to ours), we clarified methodological differences and conducted fair comparison under their settings. LearnPruner outperforms Dynamic-LLaVA by 1.5\% on average, validating the superiority of our two-stage approach. For diversity-based methods (DivPruner, DART), we added discussions in Related Work and experimental comparisons, demonstrating consistent superiority across benchmarks.
- Practical implementation and efficiency (R_ui91). We addressed concerns about FlashAttention compatibility. Since attention extraction is only required once at the k-th layer during the prefill phase, and prior work (SparseVLM) has demonstrated a dual FlashAttention approach for this purpose, the overhead is minimal. Our experiments confirm this, achieving significant inference acceleration while maintaining compatibility with modern attention mechanisms.

We also conducted comprehensive additional experiments to address specific questions:

- Data Efficiency (R_iSWN, uDiD): We demonstrated the robustness of our Learnable Pruning Module (LPM). Training with only 5% of the data yields performance comparable to using the full dataset, highlighting the module's lightweight and data-efficient nature.
- Out-of-Distribution Generalization (R_FyhA): We added evaluations on SeedBench and MMMU. LearnPruner achieves strong results on these unseen domains (e.g., charts, diagrams), proving its capability to generalize beyond training data distributions. Additionally, Video-LLaVA experiments show that LPM trained on images alone generalizes effectively to video tasks, demonstrating generalization from another perspective.
-  Ablation Studies :

    - Proportion of diverse tokens $\lambda$ (R_iSWN, R_ui91): We added ablation on the diversity-based token selection module, adjusting $\lambda$ to control the proportion of diverse tokens. We empirically set $\lambda=0.1$ to balance foreground semantics and background context.

    - Pruning Layer $k$ (R_FyhA) : We provided more comprehensive ablation experimetns on pruning positions and the results show that LearnPruner maintains consistent performance across different model architectures and benchmark tasks.
- Failure Case Analysis (R_iSWN): We added several failure cases  (Figure 7) with detailed discussion in Appendix to provide an objective analysis of our method's limitations.

We sincerely hope that our responses and supplementary experiments demonstrate the effectiveness of our method, address reviewer concerns, and facilitate the area chair's understanding of our discussion with reviewers. Moreover, we are grateful to R_ui91 for their recognition of our efforts and **the score increase from 2 to 4**, and we remain committed to further improving the manuscript based on all valuable feedback received.

Best regards,

Authors of Submission 15007

---

### Meta-Review · Area_Chair_jkLA · 2026-01-06

**Summary:**

This paper proposes a token-pruning strategy for vision language models that preserves 95% of the original performance while using 5.5% of vision tokens and achieves 3.2x inference acceleration. The paper first provides an analysis of attention in the vision encoder versus the LLM. They particularly find that CLS token attention may not effectively focus on salient foreground regions. They propose a learnable pruning strategy that feeds token features to a lightweight MLP and prune tokens based on the softmax on the output of the MLP. In a second stage text-irrelevant content is also removed.

The reviewers note strengths of the works including insightful examination of the attention behavior, novel learnable pruning module, well-motivated method, and strong performance.

**Reviewer Concerns:**

Reviewer ui91
- **Similarity and comparison to Dynamic-LLaVA**: The authors acknowledge the similarity. They note that Dynamic-LLaVA requires fine-tuning the entire base VLM while their approach trains the lightweight LPM only. The authors further include additional experiments following Dynamic-LLaVA’s training and testing settings and find that using only 25% of vision tokens their method achieves better performance than the base VLM.
- **Description of diversity-based token selection module**: The authors revised the description in Section 3.3 and included a new ablation in Appendix A.2.
- **Discussion of diversity-based vision token pruning and comparison with DivPrune and DART**: The authors added description in Section 2.2 and included experimental comparisons in Table 1 and 2.
- **Implementation efficiency and compatibility with FlashAttention, vLLM, and TensorRT**: The authors provide details for how the method can be implemented for compatibility. The reviewer emphasized that further overhead analysis is required and that modifying vLLM may not be straightforward. However, the reviewer notes that this does not influence their score. The authors provide additional experiments for the overhead for FlashAttention compatibility.

Reviewer FyhA:
- **Generalization to other tasks**: The authors provide additional evaluations on SeedBench and MMU in favor of the proposed method.
- **Compression ratio of the vision encoder**: The authors refer to the ablations provided in Appendix A.3.2.
- **Choice of hyperparameter k (second pruning stage layer)**: The authors refer to the existing ablations in the Appendix and Figure 2(c). They also provide additional results for model and benchmark sensitivity.
- **Query-agnostic pruning**: The authors note that their first stage is inherently query-agnostic and the method can operate with only the first stage which as shown in Table 6 outperforms common query-agnostic methods such as [CLS] attention-based pruning.

Reviewer uDiD:
- **Comparison using Qwen-2.5-VL**: The authors refer to the results in Appendix A.3.1-Table 6. The results are moved now to Section 4.2-Table 4 as part of the main text.
- **Novelty of “Remove Text-Irrelevant Content”**: The authors acknowledge that this part may not be novel while noting that their key innovation is in adopting appropriate pruning criteria in different stages and not the individual components.
- **Additional training is required compared with training-free methods**: The authors acknowledge this limitation but note that the training cost is modest (5% of training data) while providing substantial performance gains.

Reviewer iSWN:
- **Missing failure case analysis**: The authors point to analysis in Figure 7 and provide further discussions and details in Appendix A.3.
- **Fewer training samples**: The authors provide a new ablation study and included in Appendix A.2 for data efficiency for LLaVA-7B and LLaVA-NeXT-7B showing that their method achieves strong performance with only 5% of the training data.
- **Ablation study for diversity-based token selection**: The authors provide a new ablation study in Appendix A.2.
- **Attention bias**: The authors argue their results in Figure 2(b) and Section 3.2 show that text-to-vision attention becomes reliable and stable starting from the middle layers and attention bias does not pose a significant bottleneck.

**Reviewer Scores:**

Reviewer ui91 initially gave a score of 2 (reject) and then raised to 4 (marginally below acceptance threshold). Reviewer uDiD gave a score of 4 (marginally below acceptance threshold). Reviewers FyhA and iSWN gave a score of 6 (marginally above acceptance threshold)

The AC finds that all reviewer concerns have been addressed during the rebuttal through new results and changes to the paper. The authors are recommended to explore releasing an implementation compatible with popular inference frameworks such as vLLM or TensorRT.

---

### Decision · Program_Chairs · 2026-01-26

Accept (Poster)